# Gain of Alternative Allele Expression of *LINCO2449* at rs149707223 in Schizophrenia and Bipolar Disorder: Inducing Synaptic Transmission and Behavioral Deficits in Mice

Tengfei Yang [1,2,7], Jin-Ming Liu[1,3,7], Qiaqi Chen[1,2,7], Zhiying Deng[1,2,7], Chaoying Ni[1,2,7], Yunqian Wang[1,2], Yuting Lan[1,2], Tingyun Jiang[4], Shufen Li[1,2], Meijun Jiang[5], Hong Xue [6], Xiong Cao [1,3] ✉, Zhongju Wang [1,2] ✉ & Cunyou Zhao [1,2,5] ✉

Bipolar disorder (BD) and schizophrenia (SZ) are complex psychiatric disorders with overlapping features. Their heterogeneity may arise from interactions between genetic variants and environmental or epigenetic modifiers. Allele-specific expression (ASE), an imbalance in expression between gene alleles, provides a key mechanism linking these interactions to disease. We conducted transcriptomic and genomic analyses in phenotype-discordant monozygotic twins to investigate ASE in psychiatric risk. Nine ASE-affected long non-coding RNAs were identified, including *LINCO2449*, which showed a consistent allele-specific shift in BD/SZ patients favoring the alternative G allele at rs149707223. Functional analyses revealed that overexpression of the *LINCO2449* G allele in mice induced social deficits and repetitive behaviors. These phenotypes were associated with enhanced excitatory transmission within the mPFC-NAc circuit, mediated by the synaptic regulator CPLX1. Our findings demonstrate that ASE-driven dysregulation of *LINCO2449* contributes to synaptic and behavioral abnormalities, underscoring ASE as a potentially important regulatory mechanism in BD and SZ pathogenesis.

Bipolar disorder (BD) and schizophrenia (SZ) are among the most severe psychiatric disorders worldwide. Both are complex, polygenic mental illnesses characterized by disturbances in thought and emotion, as well as impairments in cognitive and affective processing. These disorders are the leading cause of disability[1,2]. The complexity of BD and SZ arises not only from their heterogeneous symptomatology but also from their multifactorial etiology, which involves genetic, developmental, and environmental influences. Genetic and epidemiological studies have revealed a substantial degree of overlap between BD and SZ, with heritability estimates for both disorders ranging from 60 to 80 %[3].

Recent genome-wide association studies (GWAS) have identified numerous disease-associated genetic variants, the majority of which are non-coding[4–8]. This suggests that these variants may contribute to disease susceptibility by regulating gene expression, thereby mediating interactions between genetic risk, environmental exposure, and phenotypic outcomes. Gene expression variability plays a crucial role in determining cellular function and characteristics, often manifesting as allele-specific expression (ASE), a fundamental yet understudied mechanism of gene regulation in disease. ASE arises when one allele at a given locus is preferentially expressed over the other, leading to functional consequences that may influence disease pathogenesis[4,9–11]. ASE is

a dynamic and variable phenomenon, exhibiting tissue-specific and inter-individual variability[12–16]. Since both alleles of a genetic variant exist within the same cellular or nuclear environment, significant inter-individual differences in allelic expression patterns may contribute to the heterogeneity of symptoms observed in complex psychiatric disorders.

The ASE approach provides a powerful framework for investigating the relationship between epigenetic variation and gene expression in disease. ASE signatures have been shown to be influenced by environmental factors, health conditions, and disease states[17]. ASE discordance has important implications for understanding the molecular mechanisms underlying phenotypic variability, as it may contribute to both Mendelian and complex genetic traits. For example, ASE can increase disease susceptibility by exacerbating the deleterious effects of mutations via haploinsufficiency or mosaic somatic expression[10,18–22]. Studies of epigenetic profiles in monozygotic (MZ) twins, particularly those with discordant phenotypes, provide a unique opportunity to dissect the causes and consequences of ASE. MZ twins share an identical genetic background, age, sex, cohort effects, maternal effects, and early-life environment, making them an ideal model for exploring the interplay between genetic, epigenetic, and environmental factors. Stochastic ASE switching, resulting in discordant ASE patterns among MZ co-twin pairs, suggests that ASE may contribute to individual differences in disease susceptibility and phenotypic variation.

Long noncoding RNA (lncRNA), a class of RNA molecules exceeding 200 nucleotides in length and lacking protein-coding potential, represents a significant epigenetic regulatory factor. Approximately 40% of lncRNAs are predominantly expressed in the brain, where they play crucial roles in neurodevelopment and the pathophysiology of psychiatric disorders[23–28]. LncRNAs participate in a wide range of biological processes, interacting with target DNA, RNA, or proteins via base pairing or structural domains. They regulate gene expression at both transcriptional and post-transcriptional levels through diverse mechanisms[24,29]. Moreover, functional variants within lncRNAs can influence their regulatory capacity in multiple ways[30–32], affecting their ability to bind transcription factors and target gene promoters. These findings suggest that ASE of functional variants in lncRNAs may play a role in the development of psychiatric disorders.

The objective of this study was to investigate the potential role of ASE in functional lncRNA variations in the pathogenesis of psychiatric disorders. Given their shared genetic background, MZ twins serve as an ideal model for assessing the effect of allelic expression variations on phenotypic differences and disease susceptibility. To this end, we performed whole-genome sequencing and RNA sequencing in MZ twins discordant for BD or SZ, aiming to identify disease-associated ASE-lncRNAs. To further validate the functional impact of these ASE variants, behavioral, electrophysiological, and mechanistic experiments were conducted in both cellular and mouse models. Our findings suggest that alterations in the allelic expression of functional lncRNA variants may represent a pathogenic mechanism underlying the development of polygenic psychiatric disorders.

## Results

### Gain of alternative ASE of rs149707223 in *LINC02449* links to disease susceptibility in MZ twins

To explore the potential association between allele-specific lncRNA expression (ASE-lncRNA) and disease susceptibility in MZ twin pairs exhibiting discordant phenotypes (DC twins), we recruited nine pairs of DC twins (18 individuals), including four schizophrenia-discordant (SDC) and five bipolar disorder-discordant (BDC) twin pairs (Supplementary Data 1). Genomic DNA and total RNA were extracted from peripheral blood for whole genome sequencing (WGS) and strand-specific RNA-sequencing (RNA-seq). WGS data had a depth of ~700 million 150-bp pair-end reads per sample, while RNA-seq data had ~100 million 125-bp paired-end reads per sample.

We identified 20,811 tagging single-nucleotide polymorphisms (SNPs) with heterozygous genotype in at least one twin pair, mapping to 7696 lncRNAs. Using GATK4[33], SHAPEIT2[34], and g2gtools[35], we construct individual lncRNA haplotypes from WGS data (Fig. 1A). Haplotype expression levels were then quantified using an expectation-maximization algorithm for ASE (EMASE) based on RNA-seq data and tagging SNPs[36]. After filtering out SNPs with distinct homozygous genotypes in twin pairs, we applied a Bayesian generalized additive linear mixed model to calculate Bayesian factors (*BF*) and assess allele-haplotype expression differences by calculating the expression ratio of the alternative allele-haplotype relative to the reference allele-haplotype for each of the 13,366 tagging SNPs annotated in 6,201 lncRNAs. This analysis identified 124 SNPs in 92 lncRNA transcripts (85 genes) with a significant discordant ASE pattern (*BF* > 5) between affected and unaffected DC twins. Among these, nine ASE-lncRNAs containing 15 ASE-SNPs exhibited significant SZ- or BD-related expression changes in the PsychENCODE postmortem brain RNA-seq dataset (Fig. 1B and Supplementary Data 2)[37].

Given that lncRNAs can regulate gene transcription by forming RNA: DNA triplexes, we employed Triplex Domain Finder (TDF)[38] to predict allelic effects on target gene binding. Among the aforementioned 15 ASE-SNPs, only rs149707223 (C/G) in *LINC02449* transcript 2 (ENST00000304751 or NR_120455.1) exhibited allele-specific binding affinity of 428 potential target genes (Supplementary Data 3). Of these, 15 genes exhibited genotype-dependent expression correlation with *LINC02449* in the LIBD dorsolateral prefrontal cortex (DLPFC) dataset[39] (Fig. 1C and Supplementary Data 4). Gene ontology-biological processing (GO-BP) analyses revealed significant enrichments of these genes in terms related to regulation of AMPA receptor activity, antegrade trans-synaptic signaling, chemical synaptic transmission, trans-synaptic and synaptic signaling (Fig. 1D), indicating a functional role of *LINC02449* in psychiatric disorders. Notably, the SNP rs149707223 (C/G) displayed a pronounced allele-specific shift in *LINC02449* expression across two BDC twin pairs, consistently favoring the reference C allele in unaffected individuals and the alternative G allele in affected co-twins (Fig. 1B). In line with this, *LINC02449* expression was observed to be elevated significantly in BD patients (Log$_2$FC = 0.1408, *P* = 0.002; Supplementary Data 2) and showed a non-significant trend toward upregulation in SZ patients (Log$_2$FC = 0.009, *P* = 0.770) in the PsychENCODE brain RNA-seq dataset[37]. Given these findings, we further investigate the functional impact of rs149707223(C/G) on *LINC02449* to better understand its role in psychiatric disorders.

### Allele-specific overexpression of *LINC02449* in mice induces social deficits and stereotypic behaviors associated with BD

*LINC02449* (also known as *RP11-266K4.9*, ENSG00000215241) transcript 2 (NR_120455.1) is situated on the positive strand of chromosome 12 (chr12:8235415-8238946) and consists of two exons with a total length of 795 bp (Supplementary Fig. S1A). In silico prediction (Supplementary Fig. S1B, C) and in vitro fusion of an open reading frame (ORF) with enhanced green fluorescent protein (EGFP) were conducted to assess the protein-coding potential of *LINC02449*, revealing that *LINC02449* does not code for a protein (Supplementary Fig. S1D−F). Furthermore, RNA-seq from the lncATLAS dataset[40] confirmed that *LINC02449* is predominantly expressed in the nucleus (Supplementary Fig. S1G).

To investigate the allele-specific function of *LINC02449*, we generated two recombinant adeno-associated viruses (rAAV). Each rAAV vector carried *LINC02449* transcript 2 harboring either the alternative G allele (Alt) or the reference C allele (Ref) at rs149707223, driven by the human neuron-specific synapsin I promoter. An empty AAV vector served as a control (Ctrl). Given that BD-associated upregulation of *LINC02449* expression was observed in the cortex in the PsychENCODE dataset (Supplementary Data 2)[37] and that *LINC02449* is

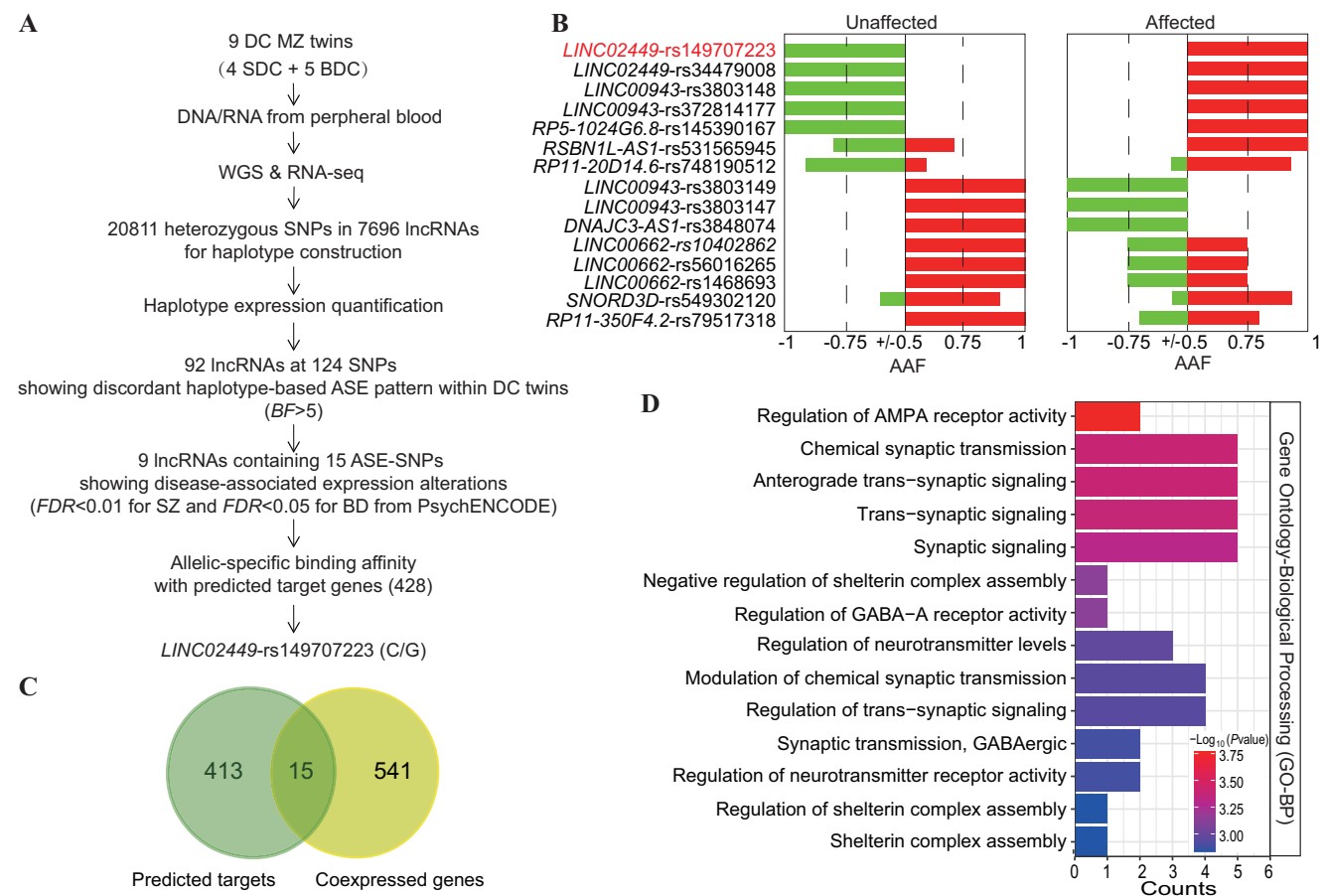

**Fig. 1 | Identification of disease-associated ASE-lncRNAs. A** Flowchart illustrating the screening strategy for disease-associated ASE-lncRNAs. **B** Allelic imbalance of the 15 ASE-lncRNA sites in affected and unaffected co-twins. Bar graphs display the fraction of individuals showing alternative allele-haplotype (AAF > 0; red) or reference allele-haplotype (AAF < 0; green) at each SNP-lncRNA pair. Data are stratified by affected or unaffected individuals. **C** Venn diagram showing the overlap between predicted *LINC02449* targets and genotype-dependent co-expressed genes in human prefrontal cortex (PFC) tissue (GTEx). **D** Gene Ontology Biological Process (GO-BP) enrichment analysis of the 15 overlapping candidate target genes of *LINC02449* was performed using ToppGene Suite. The color scale indicates the log-transformed uncorrected *P*-values for each enriched term.

highly expressed in the human cortex according to the GTEx dataset (Supplementary Fig. S2)[41], we injected rAAV into the medial prefrontal cortex (mPFC) region of 8-week-old male C57BL/6 mice (Fig. 2A). Confocal imaging confirmed accurate expression of the virus in mPFC neurons (Fig. 2B). *LINC02449* expression was successfully induced in the mice following a three-week period (Fig. 2C). Behavioral testing was conducted on day 21 after AAV injection using a battery of assays, including the three-chamber social interaction test, grooming and marble burying test, open field test, elevated plus maze, sucrose preference test, Y-Maze, forced swim test, and tail suspension test.

In the three-chamber social interaction test, mice overexpressing the *LINC02449*-alt allele (Alt) showed a reduced preference for the first introduced stranger mouse (S1) compared to Ctrl and *LINC02449*-ref allele (Ref) mice (Fig. 2D), as indicated by decreased time spent in the S1-associated chamber (Fig. 2E) and a lower preference index (Fig. 2F), suggesting impaired social interaction. However, when a second stranger mouse (S2) was introduced to the previously empty cage, all groups, including Alt, Ref, and Ctrl, exhibited a comparable preference for S2 over S1 (Fig. 2E, G), suggesting intact social novelty recognition. These findings indicate that Alt mice exhibit specific deficits in social interaction behavior without impairments in social novelty recognition.

Notably, Alt mice also displayed increased repetitive behaviors, characterized by significantly prolonged grooming duration (Fig. 2H) and a greater number of buried marbles (Fig. 2I) compared to both Ctrl mice and Ref mice. These stereotypic behaviors are commonly

associated with BD[42]. Despite *LINC02449* overexpression, Alt mice did not exhibit significant changes in hyperlocomotor activity suggestive of mania-like behavior, as indicated by total distance traveled and time spent in the center zone during the open-field test (Supplementary Fig. S3A, B), Similarly, no differences were observed in anxiety-like behavior (time spent in the open arms of the elevated plus maze; Supplementary Fig. S3C), depression-like behaviors (immobility in the forced swim test and tail suspension test; sucrose consumption in the sucrose preference test; Supplementary Fig. S3D–G), or cognitive performance in the Y-maze spatial recognition test (Supplementary Fig. S3H).

Together, these results suggest that the gain of *LINC02449* expression in mice, specifically the alternative G allele of rs149707223, selectively impairs social interaction and enhances repetitive behaviors, without affecting general locomotion, mania-like hyperactivity, anxiety-like behavior, depression-like phenotypes, or spatial cognitive function.

### *LINC02449* exhibits allele-specific enhancement of excitatory transmission in the mPFC-NAc circuit

To investigate the molecular mechanisms underlying the social interaction deficits and repetitive behaviors induced by the *LINC02449* alternative G allele in mice, we performed RNA-seq analysis of RNA extracted from the mPFC of *LINC02449*-OE mice or Ctrl mice. Principal component analysis (PCA) and clustering analysis of the expression

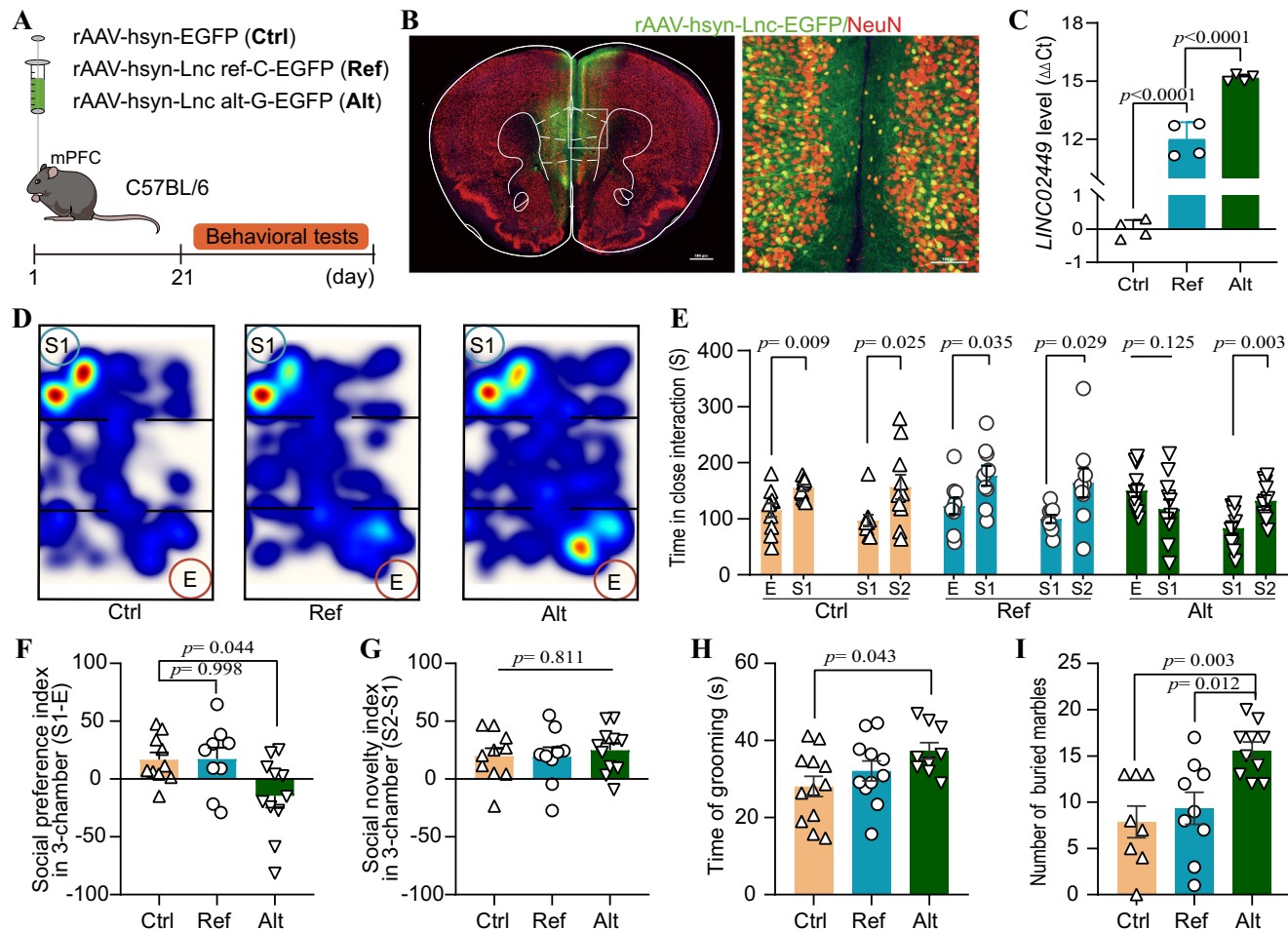

**Fig. 2 | Behavioral experiment results in mice overexpressing lncRNA**
***LINC02449.* A** Schematic of recombinant adeno-associated virus (rAAV) injection
and experimental paradigm. **B** Representative images of rAAV-hsyn-LINC02449-
EGFP expression in the medial prefrontal cortex (mPFC) of *C57BL/6 J* mice. The
green signal represents EGFP, while the red signal indicates NeuN + cells. Scale bars
represent 500 μm (left) and 100 μm (right). **C** Relative expression levels of
*LINC02449* in the mPFC of mice overexpressing the *LINC02449* reference allele (Ref;
*n* = 4) or the *LINC02449* alternative allele (Alt; *n* = 4), compared to control mice
(Ctrl; *n* = 4), shown as delta delta Ct values from four Ctrl, Ref, and Alt mice.
**D** Representative images of the three-chamber test, comparing a chamber with the
first introduced stranger mouse (S1) versus the empty cage (E). **E** Time spent in
close interaction in the three-chamber test, comparing time spent in the chamber
with the first introduced stranger mouse (S1) versus the empty cage (E) or versus
the second stranger (S2). **F** Social Preference index, indicating impaired social
desirability, derived from the numerical difference between the time spent in the
chamber with the S1 and in the empty cage (E), divided by the total time spent,
multiplied by 100. **G** Social novelty recognition, indicated by the social novelty
index (Ctrl: *n* = 10; Ref: *n* = 9; Alt: *n* = 11). **H, I** Repetitive behavior tests, shown by
grooming time (**H**; Ctrl: *n* = 12; Ref: *n* = 11; Alt: *n* = 9) and number of buried marbles
(**I**; Ctrl: *n* = 8; Ref: *n* = 9; Alt: *n* = 10). All data represent as means ± standard devia-
tions (SD). One-way ANOVA followed by Tukey's multiple comparison test was
performed to generate adjusted *P*-values for the indicated comparisons, except for
panel E, where *P*-values were calculated using a two-tailed Student's *t* test. Source
data are provided as a Source Data file.

data, utilizing Fragments Per Kilobase of exon model per Million
mapped fragments (FPKM) values for genes with an average FPKM > 1,
revealed that the gene expression pattern in the Ref mice fell between
that of the Alt mice and the Ctrl mice (Supplementary Fig. S4). As a
result, we treated *LINC02449* overexpression with the Alt allele, Ref
allele and Ctrl as ordinal categorical variables in DESeq2 analysis,
identifying 1395 differentially expressed genes (DEGs) with adjusted *P*-
values (*P*adj) < 0.05 under Benjamini-Hochberg correction to control
the false discovery rate (Fig. 3A) and Supplementary Data 5). Func-
tional enrichment analyses revealed significant association with GO-BP
such as neuron projection development and neuron development, and
GO-cellular components (GO-CC) including synapse, postsynapse,
neuron projection, and neuron to neuron synapse (Fig. 3B). Enrich-
ment for human phenotypes, including autistic behavior and abnormal
emotion/affect behavior, and mouse phenotypes, such as abnormal
synaptic physiology/transmission and abnormal emotional/affect
behavior, were also noted (Fig. 3C).

To complement these findings, we infected SK-N-SH neuro-
blastoma cells with lenti-virus (LV) carrying *LINC02449* with the Alt-G
allele or the Ref-C allele, utilizing an empty LV vector as a control. Upon
examination of the DEGs induced by *LINC02449* overexpression, we
also found that the Alt allele had a more pronounced effect on gene
expression than the Ref allele (Fig. 3D). A total of 419 DEGs (*P* < 0.05 & |
Log2FC | >0.5) were identified when considering the Alt allele, Ref
allele, and Ctrl as ordinal categorical variables in the DESeq2 analysis of
RNA-seq data (Supplementary Data 6). Functional enrichment analysis
of upregulated DEGs revealed significant associations with GO-BP
terms such as chemical synaptic transmission, trans-synaptic signaling,
and neuron projection development (Fig. 3E), and mouse phenotypes
such as abnormal synaptic transmission, abnormal synaptic physiol-
ogy, abnormal social/conspecific interaction behavior, and abnormal
nervous system electrophysiology (Fig. 3F and Supplementary Data 7).
Together, these findings suggest that *LINC02449* overexpression-
induced changes in gene expression are associated with synaptic

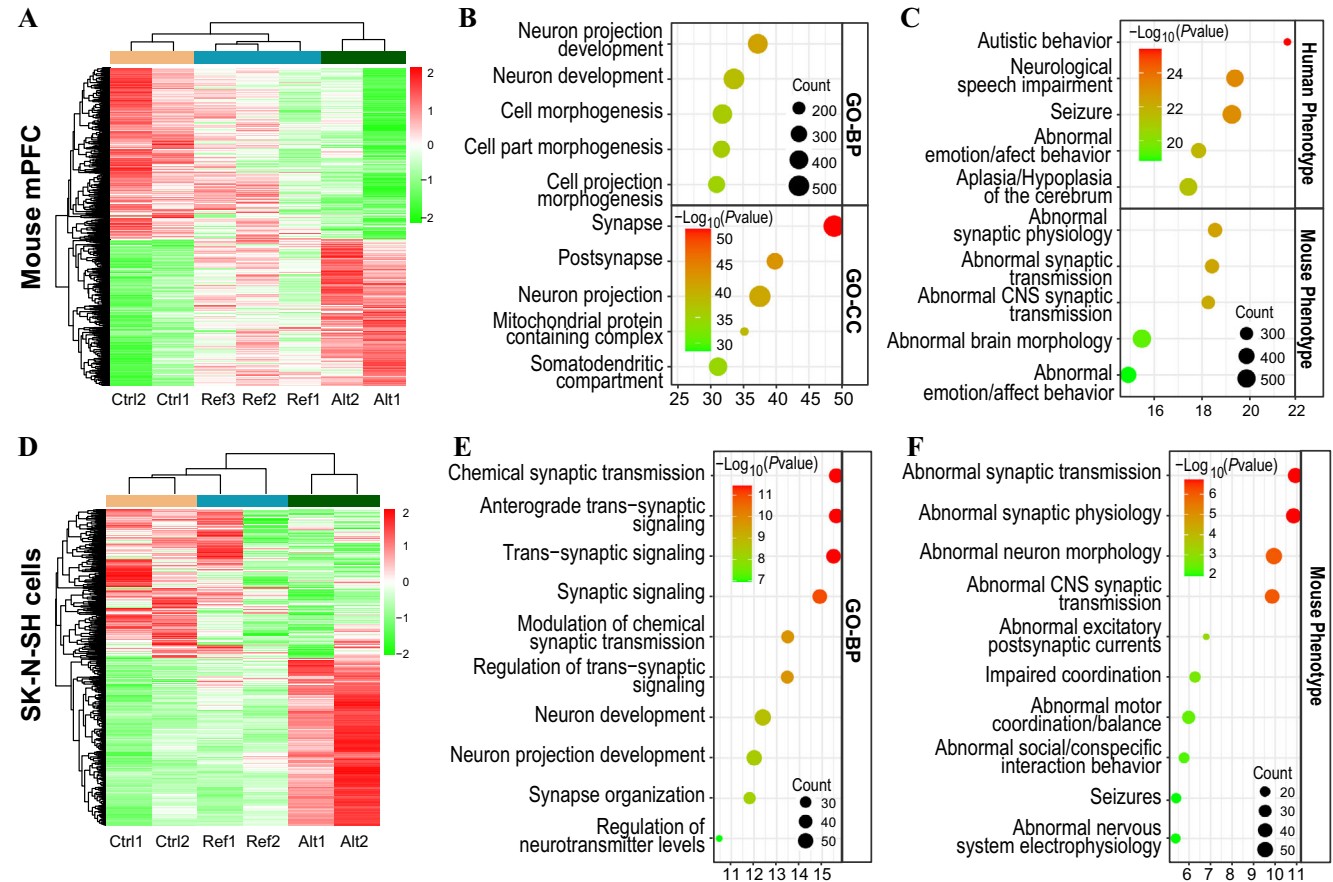

**Fig. 3 | Functional enrichment analysis of differentially expressed genes (DEGs) induced by *LINC02449* overexpression in mouse mPFC and SK-N-SH cells. A** Heat map of cluster analysis of DEGs induced by *LINC02449* overexpression (adjusted *P* < 0.05 from DESeq2) in mouse mPFC. **B**, **C** Gene Ontology (GO) biological process (GO-BP) and cellular component (GO-CC) enrichment analysis (**B**) and human and mouse phenotype enrichment analysis (**C**) of *LINC02449*

overexpression-induced DEGs from mouse mPFC. **D** Heatmap of cluster analysis of *LINC02449* overexpression-induced DEGs (*p* < 0.05 and |log₂FC| > 0.585 from DESeq₂) in SK-N-SH cells. **E**, **F** GO-BP (**E**) and mouse phenotype (**F**) enrichment analysis of *LINC02449* overexpression-induced upregulated DEGs in SK-N-SH cells. Enrichment analysis was performed using ToppFun; color intensity reflects the log-transformed uncorrected *P*-values for each enriched term.

dysfunction, which may underlie the observed social interaction deficits and repetitive behaviors.

To ascertain whether *LINC02449* overexpression leads to Alt allele-specific alterations in synaptic function, we conducted whole-cell patch clamp recordings to measure action potentials in layer IV-V pyramidal neurons in the mPFC of mice (Fig. 4A). We found that *LINC02449* overexpression had minimal influence on resting membrane potential (RMP), rheobase, and threshold voltage in mPFC pyramidal neurons (Fig. 4B–D). In addition, no significant differences were observed in the frequency or amplitude of action potentials in response to depolarizing current injection (Fig. 4E–G). Next, we examined the mPFC-nucleus accumbens (NAc) excitatory transmission circuit to assess potential aberrant excitatory transmission. The mPFC-NAc circuit maintains intact transmission structures when examined electrophysiologically in isolated brain slices, and NAc neurons that receive augmented excitatory transmission from other brain regions have been observed to induce social deficits in mice[43,44]. Using an in vivo optogenetic approach, we injected a channelrhodopsin-2 (ChR2) viral vector into the mPFC and delivered blue light stimulation to the NAc via fiber optic cannulas (Fig. 4H). In the three-chamber interaction task, blue light stimulation of the NAc resulted in reduced social interaction in ChR2-expressing mice (ChR2) compared to control mice expressing enhanced yellow fluorescent protein (EYFP-expressing, Ctrl) (Fig. 4I, J). This indicates that activation of the mPFC-NAc circuit can impair social behavior.

We also recorded miniature excitatory postsynaptic currents (mEPSCs) in NAc neurons (Fig. 4K), and found that the mEPSC frequency (Fig. 4L) and amplitude (Fig. 4M) were significantly higher in Alt neurons compared to Ref and Ctrl neurons. However, no differences were observed in neuronal complexity, dendrite length, or spine density between the Alt, Ref, and Ctrl neurons (Supplementary Fig. S5). Overall, these results demonstrate that overexpression of the *LINC02449* alt G allele in the mPFC enhances mEPSC activity in NAc neurons, suggesting that *LINC02449* plays a crucial role in regulating synaptic transmission within the mPFC-NAc circuit, potentially contributing to the observed social interaction deficits and repetitive behaviors.

**Genome-wide analysis reveals CPLX1 as one of the key downstream targets regulated by *LINC02449***

To identify potential targets regulated by the *LINC02449* Alt-G allele that contribute to synaptic dysfunction, we examined whether DEGs induced by *LINC02449* overexpression (OE) in SK-N-SH cells and mouse mPFC also exhibited coexpression with *LINC02449* in the human brain (Fig. 5A). Notably, upregulated DEGs from both the mice mPFC (*n* = 2 + 266 genes; enrichment *P* = 2.31e-12 and OR = 2.31) and SK-N-SH cells (*n* = 2 + 53 genes; *P* = 3.97e-3 and OR = 2.16) were significantly enriched for genes positively correlated with *LINC02449* expression (R > 0.3, *P* < 1.0e-5) in the GTEx human PFC brain RNA-seq dataset[41] (Supplementary Data 8). Among the overlapping two genes,

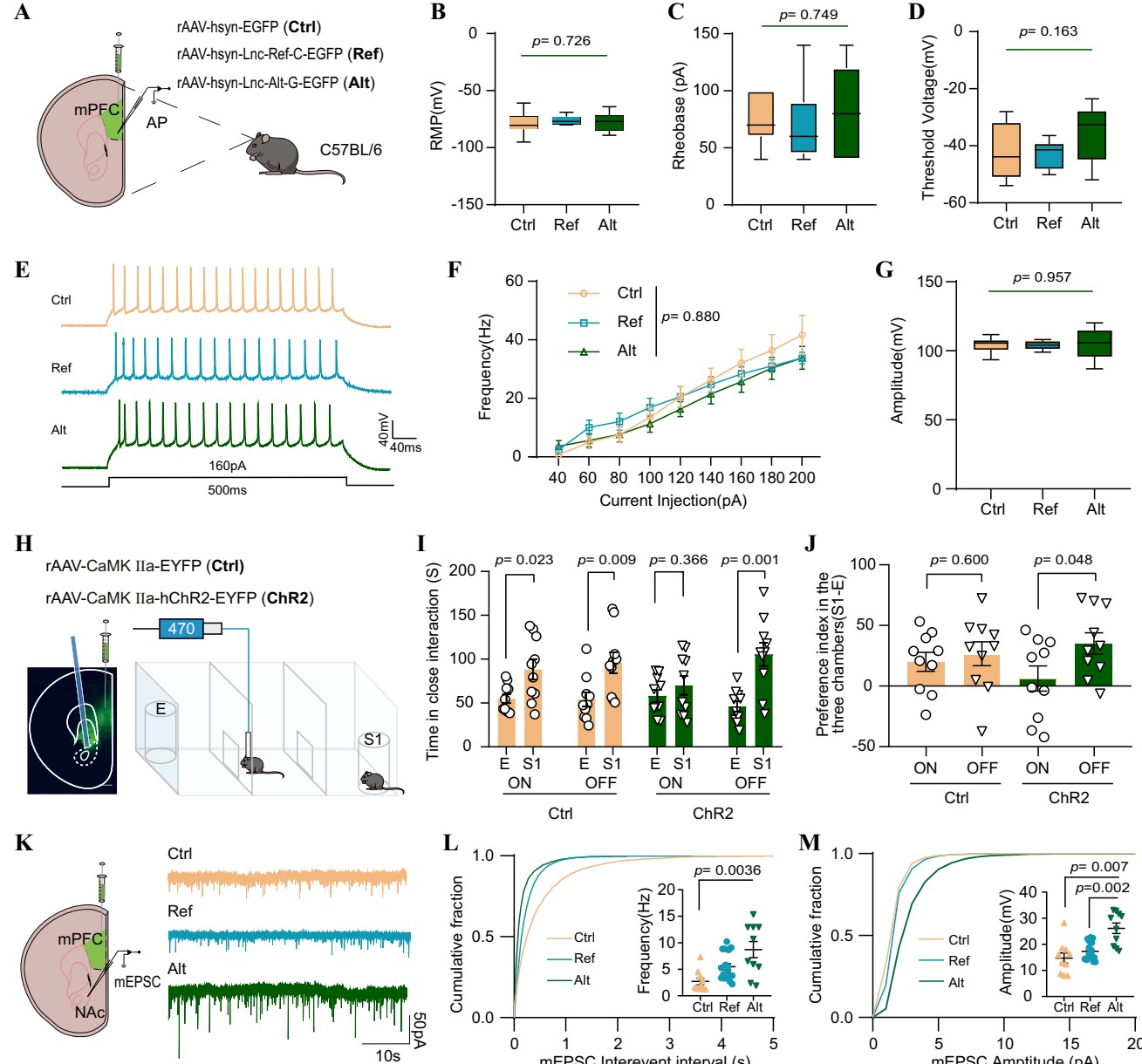

**Fig. 4 | Electrophysiological characteristics of *LINC02449*-overexpressed mice in the mPFC-nucleus accumbens (NAc) circuit. A** Schematic experimental paradigm for rAAV injection and action potential (AP) recording. **B–D** Comparable resting membrane potentials (RMPs, **B**), rheobases (**C**), and AP thresholds (**D**) in mPFC pyramidal neurons. **E** Representative traces of AP firing evoked by depolarizing currents. **F** Mean of AP firing frequency ± standard errors of the mean (SEMs) plotted against increasing current steps. **G** AP amplitude in mPFC pyramidal neurons. $n = 8$ neurons from three control mice (Ctrl); $n = 8$ neurons from three mice expressing *LINC02449* reference allele (Ref); $n = 9$ neurons from three mice expressing *LINC02449* alternative allele (Alt). The bold lines, upper boundaries, and lower boundaries of notches represent the medians, 75th percentiles, and 25th percentiles, respectively. Whiskers extend 1.5 times the interquartile range (IQR). Overall, *P*-values were calculated using one-way (panels **B**–**D** and **G**) or two-way (panel **F**) ANOVA test. **H** Schematic experimental paradigm showing the effect of optogenetic stimulation of the mPFC-NAc circuit on the social function of mice in the three-chamber interaction task. Representative images show rAAV-CaMKIIa-hChR2-EYFP expression in the mPFC (green, EYFP) and optogenetic stimulation of NAc neurons. **I, J** The effects of optogenetic stimulation of the mPFC-NAc circuit on social desirability, indicated by the time spent in close interaction (**I**) and the social preference index (**J**; Ctrl: $n = 10$; ChR2: $n = 10$). **K** Schematic experimental paradigm for rAAV injection in the mPFC and miniature excitatory postsynaptic currents (mEPSCs) recording in NAc neurons, along with representative mEPSC traces. **L, M** Cumulative probability plots of the frequencies (**L**) and amplitudes (**M**) of mEPSCs recorded in NAc neurons. $n = 10$ neurons from three Ctrl mice; $n = 20$ neurons from three Ref mice; $n = 10$ neurons from three Alt mice. Data in panels (**I, J, L**, and **M**) are presented as means ± SEM, and *P*-values were calculated using a two-tailed Student's *t* test. Source data are provided as a Source Data file.

*CPLX1* and *STMN2*, *CPLX* — which encodes complexin 1— is of particular interest due to its well-established roles in synaptic vesicle exocytosis, neurotransmitter release, and chemical synaptic transimssion[45,46]. These functions are directly relevant to the synaptic transmission deficit observed following *LINC02449* overexpression. The expression level of *CPLX1* was observed to be upregulated both in *LINC02449* OE mice (Log$_2$FC = 0.277 and *P*adj = 6.1e-4; Fig. 5A) and SK-N-SH cell lines (Log$_2$FC = 0.788 and *P*adj = 9.3e-9). In addition, a positive correlation between *LINC02449* and *CPLX1* expression was observed in the GTEx PFC data (coefficient $R = 0.463$ and $P = 1.67e-12$). Furthermore, the *LINC02449* Alt-G allele-specific increase in *CPLX1* expression – but not in *STNM2* – was confirmed at the mRNA level by qPCR in human SK-N-

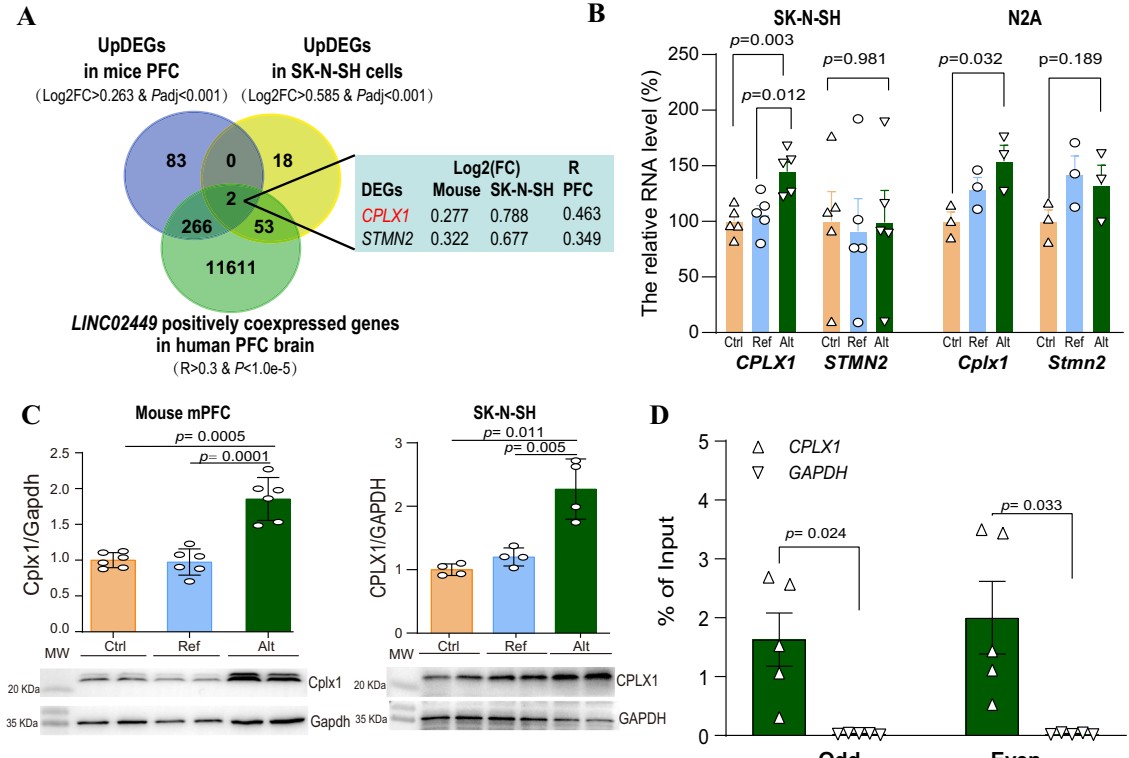

**Fig. 5 | Identification and validation of CPLX1 as a downstream target of *LINC02449*. A** Venn diagram showing two potential genes (*CPLX1* and *STMN2*) identified by intersecting DEGs induced by *LINC02449* overexpression in mouse mPFC and human SK-N-SH cells, with the *LINC02449* positively correlated genes from the human PFC from the GTEx dataset. Log₂fold-change (Log₂FC) values for each tissue and Pearson correlation coefficient (R) from GTEx data are summarized on the right. **B** qPCR analysis of *CPLX1* and *STMN2* mRNA levels in human SK-N-SH and mouse N2A cells overexpressing *LINC02449* reference (Ref) or alternative (Alt) alleles. *CPLX1*, but not *STMN1*, was significantly upregulated by the Alt allele. Data are normalized to the Ctrl group (set to 100%). **C** Western blotting validation of CPLX1 protein levels in mouse mPFC and SK-N-SH cells following *LINC02449* overexpression, with molecular weight (MW) indicated. Protein expression was normalized to GAPDH. Representative blots are shown below the quantification. **D** Chromatin isolation by RNA purification (ChIRP)-qPCR analysis showing enrichment of *LINC02449* binding at the *CPLX1* upstream region in SK-N-SH cells overexpressing the alternative allele (Alt). The retrieved *CPLX1* DNA from the *LINC02449* ChIRP analysis was quantified by qPCR and is expressed as the percentage of input in columns. The *GAPDH* (▽) served as a negative control. Odd: odd-numbered probes (#1, 3, 5); Even: even-numbered probes (#2, 4, 6). *P*-values were calculated using a two-tailed unpaired Student's *t* test (panels **B** and **C**) or a paired Student's *t* test (panel **D**). Data are means ± SEM from ≥ 3 independent triplicate experiments. Source data are provided as a Source Data file.

SH and mouse N2A cells (Fig. 5B), and the corresponding increase in *CPLX1* protein was validated by Western blotting in both SK-N-SH cells and mouse brain tissue (Fig. 5C). Based on this converging evidence, *CPLX1* was prioritized as the primary downstream effector for subsequent functional validation.

LncRNAs can regulate gene transcription by forming RNA:DNA triplex structures through direct binding to genomic DNA. Using LongTarget[47], we predicted several potential *LINC02449* binding sites within the *CPLX1* upstream region (Fig. S6). To validate this interaction, we performed a chromatin isolation by RNA purification (ChIRP) assay followed by qPCR analysis of DNA retrieved from *LINC02449*-ChIRP. The results confirmed that *LINC02449* enrichments at the predicted region spanning from 5,572-5,650 bp upstream of the *CPLX1* transcription start site (Fig. 5D), supporting a direct regulatory role of *LINC02449* in *CPLX1* transcription.

To assess the function relevance of CPLX1 regulated by *LINC02449*, we performed a neuronal vesicle release assay using FM4-64 dye in *LINC02449*-overexpressing SH-SY5Y and SK-N-SH cell lines. Following depolarization with 75 mM KCl, cells overexpressing the Alt allele exhibited a significantly faster decline in FM4-64 fluorescence intensity compared to Ref or Ctrl cells (Fig. 6A, B), indicating enhanced vesicle exocytosis. Notably, this accelerated dye loss was significantly attenuated upon *CPLX1* knockdown in SK-N-SH cells (Fig. 6C), supporting the role of CPLX1 as a key mediator of *LINC02449*-induced vesicle exocytosis.

To explored whether Cplx1 knockdown could rescue the behavioral and synaptic abnormalities associated with *LINC02449* alt-allele in mice, we injected a rAAV carrying Cplx1-shRNA into the mPFC regions of *LINC02449* alt-allele OE mice (Fig. 7A, B). This intervention led to a significant reduction in Cplx1 expression (Fig. 7C, D and Supplementary Fig. S7). Behavioral testing revealed that knockdown of *Cplx1* in *LINC02449* alt-allele OE mice (Cplx-rescue mice, RE) ameliorates the observed social and stereotypic behaviors (Fig. 7E–H). Furthermore, knockdown of *Cplx1* in RE mice normalized the frequency and amplitude of mEPSCs in NAc neurons, bringing them to levels comparable to those observed in Ctrl mice (Fig. 7I–K). These results demonstrated that the aberrant social behavior and repetitive actions observed in *LINC02449* alter-allele OE mice are alleviated by restoring normal excitatory synaptic transmission in the mPFC-NAc circuit through the inhibition of Cplx1.

## Discussion
This study provides a comprehensive investigation of ASE patterns of lncRNAs in phenotype-discordant MZ twins, highlighting their association between ASE alterations and psychiatric disease susceptibility. We identify distinct ASE patterns in lncRNA in individuals with BD or SZ, compared to healthy controls within MZ twin pairs, underscoring the role of ASE alterations in the pathogenesis of BD and SZ. Notably, the SNP rs149707223 shows a pronounced allele-specific shift in *LINC02449* expression, with a preference for the alternative allele in

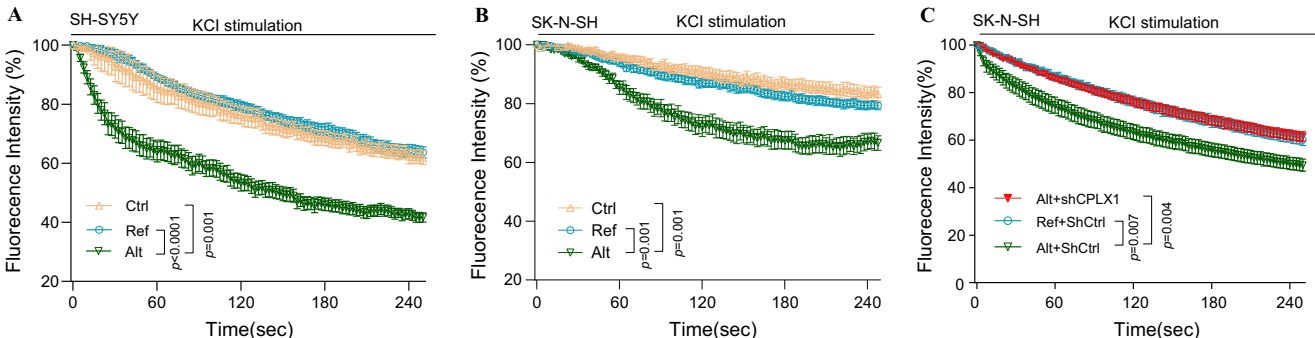

**Fig. 6 | FM4-64 imaging reveals vesicle release deficit associated with the *LINC02449* alternative allele and CPLX1 dependence. A**, **B** FM4-64 fluorescence decay following KCl-induced depolarization in SH-SY5Y cells (**A**) overexpressing control vector (Ctrl; *n* = 8 cells), *LINC02449* reference allele (Ref; *n* = 5 cells), or *LINC02449* alternative allele (Alt; *n* = 7 cells), and in SK-N-SH cells (**B**; Ctrl: *n* = 5; Ref: *n* = 7; Alt: *n* = 3) for replication. **C** Rescue experiment in SK-N-SH cells showing that knockdown of *CPLX1* (shCPLX1) attenuates the enhanced vesicle release phenotype induced by the *LINC02449* Alt allele. Ref group was co-transfected with control shRNA (Ref+shCtrl; *n* = 10), while the Alt group was co-transfected with either shCtrl (Alt + shCtrl; *n* = 14) or shCPLX1(Alt + shCPLX1; *n* = 11). Data are means ± SEMs. *P*-values were calculated using the Wilcoxon rank-sum and signed-rank tests. Source data are provided as a Source Data file.

affected individuals. This shift in expression correlates with social interaction deficits, repetitive behaviors, and enhanced excitatory transmission in the mPFC-NAc circuit in mice, mediated in part by dysregulation of the *LINC02449*–CPLX1 axis. Our findings highlight the importance of ASE in modulating gene expression and its potential contribution to the molecular underpinnings of psychiatric disease.

We demonstrate that ASE variants in lncRNAs, such as *LINC02449*, may contribute to psychiatric disease susceptibility by regulating genes involved in critical neuronal processes, including neurotransmitter signaling and synaptic transmission. This finding aligns with growing evidence that ASE can act as a modifier of genetic risk in complex disorders. Previous studies have demonstrated that ASE patterns can vary across tissue, cell types, and individuals, with functional ASE variants potentially mediating inter-individual differences in disease vulnerability[48]. Our study supports this concept by identifying 124 ASE-SNPs across 96 lncRNAs that showed discordant ASE patterns across MZ twins. Among these, 9.8% of ASE-associated lncRNAs also exhibited differential expression linked to BD or SZ in PsychENCODE brain RNA-seq datasets, suggesting functional relevance in these contexts. Recent studies have shown that SNPs within lncRNA loci can influence various aspects of lncRNA biology, including expression, subcellular localization, RNA stability, and interaction with chromatin-modifying complex or RNA-binding proteins[26,30–32]. For example, SNPs in *MIAT* and *LINC-PINT* have been linked to altered expression and splicing patterns in neuropsychiatric conditions[26,49]. These findings align with our observation that that regulatory ASE-SNP within lncRNAs, exemplified by *LINC02449*, can significantly shape that gene regulatory networks critical for neuronal processes and neuropsychiatric disease phenotypes.

Our study also sheds light on the epigenetic and pathophysiological mechanisms by which the gain of the *LINC02449* alternative allele leads to synaptic and behavioral deficits in mice. The ASE SNP rs149707223 demonstrates a clear allele-specific conversion in *LINC02449* expression in BDC twin pairs, with the alternative allele predominating in affected individuals, contrasting with the usual pattern of reference allele predominance in unaffected individuals. Furthermore, *LINC02449* expression is upregulated in BD patients and shows a tendency for upregulation in SZ patients, as seen in the PsychENCODE brain RNA-seq dataset[37], supporting the association between dysregulated *LINC02449* expression and BD. In mice, gain of *LINC02449* expression in the mPFC region results in social deficits, repetitive behaviors, and enhanced excitatory transmission at synapse within the mPFC-NAc circuit. No significant effects of *LINC02449* overexpression were observed on anxiety-like behavior, mania-like activity, depressive-like behavior, or spatial working memory. Notably,

social deficits and repetitive behaviors—both prominent in our model—are hallmark features commonly associated with BD and SZ[42,50]. A previous study showed that *Disrupted-in-schizophrenia 1* (*Disc1*)-deficient mice exhibit cognitive impairment along with repetitive and compulsive-like behaviors[51], further supporting the functional relevance of *LINC02449* in modulating synaptic activity and neurobehavioral phenotypes. In addition, the behavioral abnormalities observed in mice are supported by GO functional enrichment analysis of DEGs induced by *LINC02449* overexpression in both mice mPFC tissue and SK-N-SH cells. Consistent with previous reports[44], we also show that activation of the mPFC-NAc circuit impairs social behavior, and that *LINC02449* overexpression in the mPFC enhances the frequency and amplitude of mEPSC in NAc neurons. Together, these findings demonstrate that *LINC02449* plays a pivotal role in regulating excitatory synaptic transmission within the mPFC-NAc circuit, and that ASE-mediated dysregulation of *LINC02449* may contribute to the social deficits observed in BD or SZ.

We acknowledge that the behavioral assays employed in this study do not fully capture the clinical heterogeneity observed in SZ and BD. Specifically, paradigms designed to assess mania-like states (e.g., amphetamine-induced hyperlocomotion) or psychosis-like behaviors (e.g., latent inhibition, prepulse inhibition under pharmacological challenge) were not included. Incorporating a more comprehensive behavioral battery in future studies will be essential to thoroughly evaluate the spectrum of SZ/BD-related phenotypes. Moreover, while our results highlight the critical involvement of the mPFC–NAc circuit in mediating the observed behavioral and synaptic alterations, we recognize that other neural regions and circuits implicated in SZ and BD pathophysiology, such as the hippocampus, amygdala, and thalamus, were not assessed. Future investigations utilizing region-specific manipulations and comprehensive circuit mapping will be important to clarify whether *LINC02449* influences broader neural networks relevant to psychiatric disorders.

Impairments in synaptic function are linked to disruption in molecules responsible for learning, memory, cognition, and other brain functions, and are often associated with social deficits and repetitive behaviors. In this context, CPLX1 (complexin 1) emerged as a putative effector gene regulated by *LINC02449* through integrative analysis of DEGs identified from RNA-seq data after *LINC02449* overexpression in both mouse mPFC and SK-N-SH cells, along with co-expression network analyses from GTEx human PFC brain RNA-seq datasets. *CPLX1* expression was positively correlated *LINC024449* expression in human PFC, and significantly upregulated following *LINC02449* overexpression in mouse mPFC and SK-N-SH cells. ChIRP assay further demonstrated a direct molecular interaction between

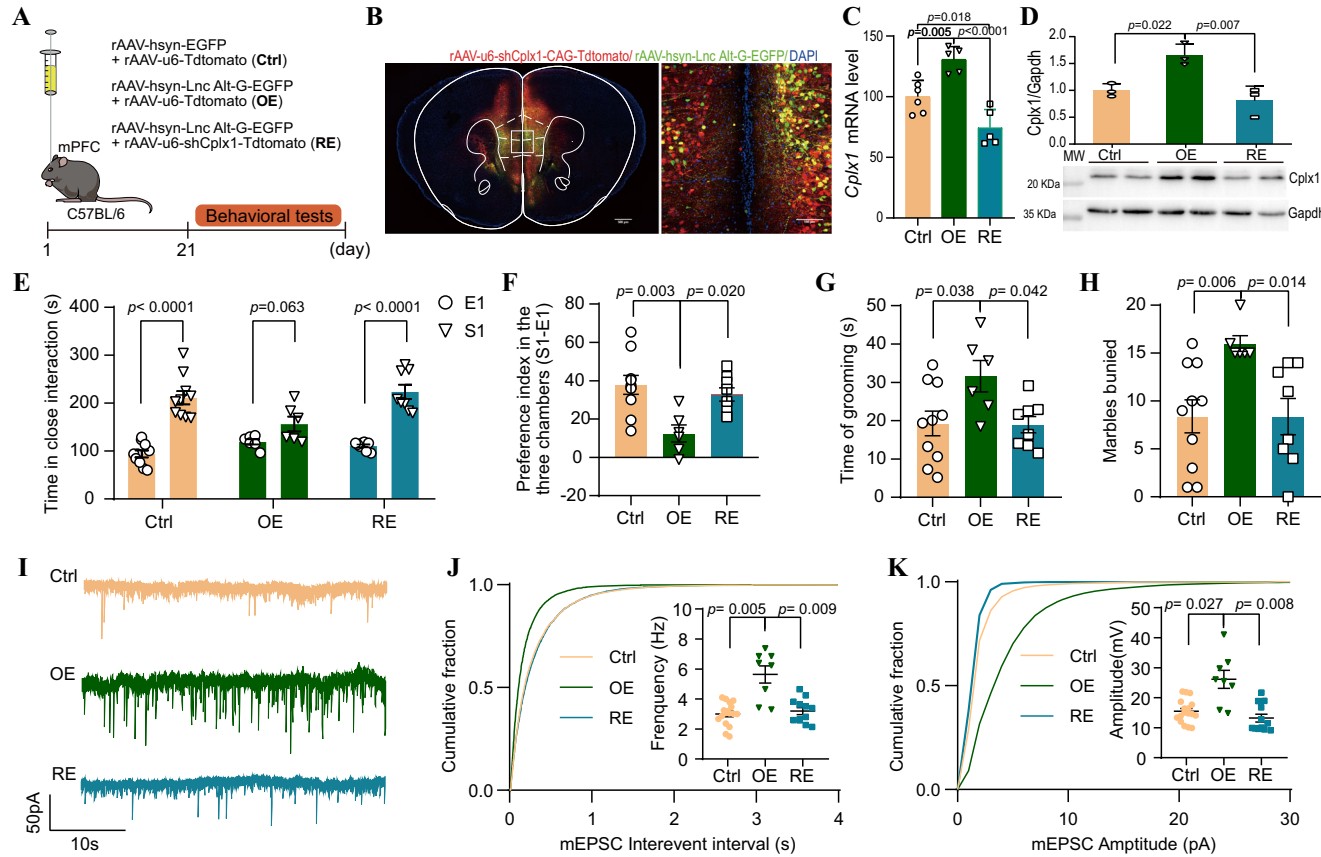

**Fig. 7 | Knockdown of Cplx1 expression ameliorates behavioral and synaptic deficits induced by the gain of *LINCO2449* alternative G allele in mice.**
**A–D** rAAVs carrying the *LINCO2449* alternative G allele (rAAV-hsyn-*LINCO2449* alt-G-EGFP, OE), *Cplx1*-shRNA (rAAV-u6-shCPLX1-Tdtomato for rescue of *Cplx1* expression in OE mice, RE), or rAAV control vectors (rAAV-u6-Tdtomato, Ctrl) were cloned into the rAAV expression vector and injected into the mPFC regions of C57BL/6 wild-type mice (**A**). Expression levels were examined by immunofluorescence analysis of brain slices (**B**), qRT-PCR analysis of *Cplx1* mRNA levels (**C**), and Western blotting analysis of Cplx1 protein levels, with molecular weight (MW) indicated (**D**). The timeline for each behavior test is indicated. Green signal represents EGFP, while red signal indicates Tdtomato cells. Scale bars represent 500 μm (left) and 100 μm (right). Data from six Ctrl, five OE, and five RE mice. **E–H** Social desirability is

indicated by the time spent in close interaction (**E**) and the social preference index (**F**), and repetitive behavior tests are shown by grooming time (**G**) and number of buried marbles (**H**) in Ctrl (*n* = 10), OE (*n* = 6), and RE (*n* = 8) mice.
**I–K** Representative traces (**I**) and cumulative probability plots of the frequencies (**J**) and amplitudes (**K**) of mEPSCs recorded in the NAc neurons from Ctrl mice (17 neurons), OE mice (8 neurons), and RE mice (12 neurons). Data are means ± SEMs. Adjusted *P*-values were calculated using one-way ANOVA followed by Tukey's multiple comparison test (panels **C**, **D**, **F**, and **G**), or Brown-Forsythe and Welch's ANOVA test followed by Dunnett's T3 multiple comparisons test (panels **H**, **J**, and **K**), except for panel (**E**), where *P*-values were calculated using a two-tailed Student's *t* test, for the indicated comparisons. Source data are provided as a Source Data file.

*LINCO2449* and the *CPLX1* upstream region, supporting that CPLX1 is a direct transcriptional target of *LINCO2449*. CPLX1 is known to regulate synaptic vesicle exocytosis, neurotransmitter release and chemical synaptic transmission[52], aligning with our findings of increased excitatory transmission in the mPFC-NAc circuit in *LINCO2449*-alt allele overexpressed mice and enhanced exocytosis process in SK-N-SH cells. Notably, knockdown of CPLX1 reversed both the synaptic and behavioral abnormalities, including social deficits and repetitive behaviors, induced by the gain of the *LINCO2449* alternative allele in mice. Previous studies have indicated that elevated CPLX1 protein levels in the orbitofrontal anterior and cingulate cortices of individuals with SZ[52], while other studies describe reduced CPLX1 levels in SZ, BD, and major depression[53,54]. These discrepancies suggest that precise CPLX1 expression levels are critical for synaptic homeostasis, potentially regulated via ASE-driven changes in *LINCO2449*.

Although CPLX1 was prioritized for functional rescue experiments due to its robust differential expression, positive correlation and direct interaction with *LINCO2449*, as well as its well-established roles in synaptic regulation, we acknowledge that it may not be the sole relevant effector gene. Knockdown of *Cplx1* could broadly affect synaptic transmission, potentially masking contributions of other genes co-

expressed with *LINCO2449*. For instance, *STMN2* identified in Fig. 5A, exhibited consistent upregulation in RNA-seq analyses, although the increase did not reach statistical significance in qPCR validation. Notably, S*TMN2* has previously been implicated in neuronal growth regulation and Alzheimer's disease pathogenesis[55,56]. Thus, future studies are warranted to explore whether targeted manipulation of STMN2— individually or in combinatorial with other candidate gene— can rescues the synaptic and behavioral phenotypes observed in our model.

Our study implicates ASE in the pathogenesis of SZ and BD, particularly through the dysregulation of the *LINCO2449*−CPLX1 axis. While CPLX1, a synaptic vesicle exocytosis regulator, represents a more tractable target, its essential role in neurotransmission raises concerns about off-target effects. In contrast, nuclear lncRNAs like *LINCO2449* are difficult to modulate due to their subcellular localization, structural ambiguity, and context-specific functions. Effective therapeutic targeting will require cell-type- and circuit-specific tools, potentially combined with allele-specific strategies to ensure precision and minimize adverse effects. From a diagnostic perspective, ASE patterns are often dynamic and tissue-specific, which can limit their utility as stable biomarkers. However, the consistent ASE shift favoring

the *LINCO2449* Alt-G allele across SZ/BD-discordant MZ twin pairs–and its dysregulation in postmortem brain tissue–suggests that stable *cis*-regulatory ASE events may serve as informative biomarkers, particularly when integrated with transcriptomic and polygenic risk data. Despite potential influences from environmental or treatment, the concordant ASE pattern among genetically identical twins raised in similar early-life environments supports a genetically driven, disease-associated mechanism. Its presence in both SZ and BD further underscores a shared molecular pathology involving synaptic dysfunction. Together, these findings position *LINCO2449* ASE dysregulation as a convergent mechanism underlying SZ and BD, with implications for disease modeling precision diagnostics.

Finally, due to the exploratory nature and limited availability of discordant MZ twin pairs, formal power calculations were not performed prior to data collection. Instead, we applied a stringent Bayesian factor (*BF* > 5) threshold within a generalized additive linear mixed model framework to robustly identify ASE differences while minimizing false positives. This approach incorporates twin pairing as a random effect and adjusts for overdispersion in allelic counts. Nevertheless, we recognize that a larger sample size could enhance the discovery of additional ASE-SNPs and improve the robustness of findings. Given the known heterogeneity in SZ and BD, expanding the cohort would likely uncover more condition-specific ASE events, reduce the risk errors, and improve the generalizability of the results. Future efforts should focus on validating these ASE patterns in larger case-control cohorts and integrating other omics layers, such as chromatin accessibility or methylation status, to refine the biological relevance of identified ASE-SNPs.

In conclusion, we identify distinct patterns of ASE of lncRNA in individuals with BD or SZ compared to healthy controls within MZ twin pairs, demonstrating a strong association between ASE alterations and disease susceptibility. Our study elucidates the epigenetic and pathophysiological mechanisms by which the gain of the *LINCO2449* alternative allele contributes to social deficits and repetitive behaviors, primarily through CPLX1-mediated enhancement of synaptic transmission. The ASE shift favoring the alternative allele in BD and SZ induces behavioral and synaptic abnormalities in mice, underscoring the potential significance of ASE as an underappreciated regulatory mechanism in the pathogenesis of major psychiatric disorders.

## Methods

### Twin subjects

In this study, we recruited nine pairs of MZ twins discordant affected by psychiatric disorders (PDC), including four pairs discordant for SZ (SDC) and five pairs discordant for BD (BDC), as detailed in Supplementary Data 1. Zygosity was determined using the Qiagen Investigator Argus X-12 QS Kit (Qiagen, USA). All participants met the diagnostic criteria for SZ or BD as defined in the fifth edition of the Diagnostic and Statistical Manual of Mental Disorders (DSM-V). Prior to participation, all individuals provided written informed consent after receiving a comprehensive explanation of the study procedures. The study was approved by the Medical Ethics Committee of Zhujiang Hospital of Southern Medical University (#2022-KY-086), Guangdong Provincial People's Hospital (#KY2024-915-02), and the third People's Hospital of Zhongshan (SSYLL20210301) and conducted in accordance with the Declaration of Helsinki. Whole-genome and transcriptome sequencing were performed using genomic DNA and RNA isolated from peripheral blood samples via the conventional phenol/chloroform method.

### Whole genome sequencing, SNP identification and annotation

To obtain comprehensive genome-wide single-nucleotide polymorphism (SNP) genotyping data, whole-genome sequencing (WGS) was performed on DNA samples from nine unaffected monozygotic (MZ) twin pairs. Approximately 700 million 150-bp paired-end reads were generated per individual sample using the Illumina NovaSeq platform, with sequencing services provided by Novogene (Tianjin, China). All sequencing data passed initial quality control assessments for base composition using FASTQC and were subsequently aligned to the human reference genome (hg19) using the Burrows-Wheeler Aligner (BWA)[57]. To remove duplicate reads, the Java-based command-line utility Picard was utilized. Base quality score recalibration of Binary Alignment Map (BAM) files was performed using the BaseRecalibrator and ApplyBQSR functions in Genome Analysis Toolkit 4 (GATK4)[33], incorporating publicly available mutation datasets (1000G_phase1, Mills_and_1000G_gold_standard.indel, dbSNP_146). Genotyping was conducted using the HaplotypeCaller module in GATK4, and variant refinement was carried out via a cross-validation process employing VariantRecalibrator and ApplyVQSR to determine the final genotype for each SNP.

SNP annotation was performed using Annovar[58], enabling the identification of SNPs located within the exonic regions of long non-coding RNA (lncRNA) transcripts. SNP-lncRNA transcript pairs were then constructed by integrating MZ twin genotype data with SNP annotation files, aligning them according to the SNP-lncRNA associations. A total of 11,773 lncRNA transcripts were identified across the genome, of which 7697 contained at least one SNP exhibiting heterozygosity in at least one pair among the nine discordant PDC MZ twin pairs. This analysis resulted in the identification of 20,813 SNPs and the formation of 25,835 SNP-lncRNA transcript pairs.

### Haplotype construction and quantification at isoform level

Leveraging the genotype files, our primary analytic step involved using SHAPEIT2[34] software to construct haplotypes for the MZ twin sets. The reference haplotype data were primarily sourced from the Chinese Han population as part of the 1000 Genome Project[59]. Next, we utilized g2gtools and vcf2doploid[35] software to modify the reference genome sequence according to the individualized haplotype information, enabling the creation of personalized haplotype reference genomes. As a result, each MZ twin pair had two sets of reference genome sequences corresponding to different haplotypes.

In the subsequent process, we conducted quality control of paired-end RNA sequencing (RNA-Seq) data from nine PDC twin pairs, yielding data from a total of 18 individuals, using FASTQC. Sequence reads were then aligned to the pair of reference genome sequences correspond to each individual using Bowtie2 algorithm[60]. After alignment, we applied the Expectation-Maximization algorithm for Allele-Specific Expression (EMASE)[36], which is specifically designed for haplotype quantification. The resulting TPM values from EMASE were then used for a comprehensive ASE analysis.

### Identification of psychiatry associated ASE

For each PDC MZ twin pair, since both the affected and unaffected individuals share identical personalized reference genome sequences, we integrated the TPM values of the two haplotypes at transcript level, as derived from the EMASE quantification results. We then examined the genotype data for SNPs located within lncRNA exons and integrate the haplotype-based TPM values across all 18 test subjects from the 9 twin pairs according to the genotype for SNPs in lncRNAs. SNPs were excluded if at least one PDC MZ twin pair showed a homozygous reference allele (Ref) and the other a homozygous alternate allele (Alt), resulting in a total of 13,368 SNPs. These SNPs corresponded to 6201 transcripts, forming 16,626 SNP-lncRNA pairings.

To further identify SNP-lncRNA transcript pairs exhibiting psychiatry-associated ASE transitions in the MZ twins, we performed a Bayesian analysis using the Integrated Nested Laplace Approximations (INLA) tool[61,62] in the R statistical programming language. Two models were formulated: Model 1, which incorporate phenotype as a fixed effect, and Model 0, a reduced model omitting the phenotype parameter. In this context, 'pis' represents the proportion of reads supporting the alternative allele in twin 'i' and disease status 's', while 'γi'

denotes the random effect corresponding to the twin.

$$M1 : logit(pis) = \beta0 + \beta s + \gamma i \qquad (1)$$

$$M0 : logit(pis) = \beta0 + \gamma i \qquad (2)$$

A likelihood ratio test comparing Model 1 and Model 0 were used to calculate Bayesian factors (*BF*). The *BF* assess the preference between the two models in the context of the ASE analysis. A *BF* value greater than 3, generally favors Model 1, indicating a potential association between the phenotype and ASE. However, in this study, we applied a more stringent threshold of *BF* > 5 to identify psychiatry-related ASE transitions, which lead to the identification of 130 SNP-transcript pairings, corresponding to 124 SNPs and 92 annotated transcripts.

## Prediction of genotype-dependent lncRNA:DNA binding

The Triplex Domain Finder (TDF)[38] is a tool used to assess the potential of RNA to bind DNA, forming triplex complexes. For lncRNAs, binding to specific DNA sequences suggests their potential to target and regulate particular genes. To investigate the impact of different alleles on DNA binding by lncRNA, we predicted the DNA sequences to which lncRNAs with different alleles would bind and compared the allele-specific differences. For SNP-lncRNA pairs identified through *BF* and filtered by PsychEncode DEG data (9 lncRNAs, 15 SNPs)[37], we extracted the sequence of the lncRNAs and manually substituted the SNP-corresponding sequence with the alternate (Alt) allele. This allowed us to obtain lncRNA sequences for both alleles. We then use the 'pro-motertest' function of TDF to analyz the global genomic promoter binding potential of each lncRNA with a specific allele, setting the '-organism' parameter to 'hg19' and keeping other parameters at their default values. By comparing the predicted promoter DNA binding profiles of different lncRNA alleles, we identified target genes with allele-specific lncRNA binding. We performed Gene Ontology enrichment analyses of these gene sets using the web-based ToppGene Suite[63].

## rs149707223-G dependent co-expression genes for *LINC02449*

To conduct our analysis, we utilized population data from the BrainSeq project[39]. First, we obtained secured genotype data for rs149707223 through the Database of Genotypes and Phenotypes (dbGaP Accession: phs000979.v3.p2). In addition, we acquired quantitative expression data from the BrainSeq website, which included expression data for *LINC02449* and all coding genes. Based on the genotype of rs149707223, we divided the samples into two groups: the wild-type (WT) group, which consisted of homozygotes for the reference C allele, and the heterozygote (Heter) group. For each group, we performed linear correlation and regression analyses between *LINC02449* and each coding gene. We defined genes in the Heter group as expression-associated if the absolute correlation coefficient r greater than 0.5 and the false discovery rate (FDR) was under 0.05. We further screened for genes whose regression coefficient (beta) in the Heter group was at least twice that of the WT group, excluding genes in the WT group with an absolute r greater than 0.5. These genes were considered to be expression-associated genes dependent on the G allele.

## DEG analysis for rs149707223 allele-specific overexpression of *LINC02449*

We performed DEG analysis by overexpressing *LINC02449* in a human neuroblastoma cell line (SK-N-SH) and in murine brain tissue. Both in SK-N-SH cells and murine brain tissue, we overexpressed the rs149707223 reference allele of *LINC02449* (ref group) and the alternate allele of *LINC02449* (Alt group), along with a control, each comprising 2 or 3 biological replicates. RNA was extracted from the

corresponding cells or brain tissue samples and subjected to transcriptome sequencing. The raw data underwent quality control via FASTQC, with low-quality reads, adapters, and base trimmed using Trimmomatic[64] to obtain clean data. The sequenced data for both human and mouse samples were aligned to the hg19 or mm38 reference genomes, respectively, using Hisat2. Quantification was performed using Stringtie and FeatureCounts[65–67].

Subsequently, the quantification results from both human and mouse samples were subjected to Principal Component Analysis (PCA) and differential analysis. The PCA indicated that the Ref group was positioned between the control and Alt groups. Based on prior correlation analysis from BrainSeq, which showed that the WT group exhibited associated genes with a regression coefficient lower than that of the Heter group, we hypothesize that *LINC02449* containing the ref allele has regulatory functionality, but to a lesser extent than the lncRNA containing the Alt allele. Based on this observation, we classified Ctrl, Ref, and Alt groups as ordered categorical variables and performed differential analysis using DESeq2[68].

## Cell culture, plasmid constructs, and transfection

Human neuroblastoma SK-N-SH (HTB-11), SH-SY5Y(CRL-2266) and HEK293T (CRL-11268) cells, and mouse neuroblastoma N2a cells (CCL-131), were obtained from ATCC. Cells were cultured in Dulbecco's Minimal Essential Medium (DMEM, Life Technologies, USA) supplemented with 10% fetal bovine serum and maintained at 37 °C in a humidified incubator with 5% CO2. Full-length lncRNA *LINC02449* cDNA (ENST00000304751) was amplified from peripheral blood RNA and cloned into the pcDNA3.1(+) expression vector (pcDNA3.1-Lnc-Ref-C). The C base at the rs149707223(C/G) site on pcDNA3.1-Lnc-ref-C was replaced by the G base via a single-point mutation, creating pcDNA3.1-Lnc-Alt-G.

To knock down *CPLX1* (ENST00000304062), a specific shRNA sequence (human shCPLX1: CCGTGTTCACTTCTAAACTAA; mouse shCplx1: GCAGCCATTGTTCTTCATATT) was designed, synthesized by Sangon Biotech (Shanghai, China), annealed to form double-stranded DNA, and cloned into the pLKO.1 vector (pLKO.1-sh*CPLX1*) using AgeI and EcoRI restriction sites. As a negative control, we used the pLKO.1-TRC control vector containing a non-hairpin insert (shCtrl: CAACAA-GATGAAGAGCACCAA), which has been widely validated as non-targeting control[69,70].

To confirm the specificity of our mouse *Cplx1*-targeting shRNA, we performed BLAST alignment against the NCBI nucleotide database. The sh*Cplx1* sequence showed 100% complementarity to *Cplx1* mRNA, with no other transcripts showing more than 80% identity. To further exclude potential off-target effects, we identified the highest-homology transcripts and assessed their expression levels by qPCR following shRNA transfection in mouse neuroblastoma (N2a) cell lines. Only *Cplx1* expression was significantly reduced, while the expression of potential off-target genes remained unchanged, indicating high target specificity (Supplementary Fig. S7).

After transfecting the specified plasmids (pcDNA3.1, pcDNA3.1-Lnc-Ref-C, pcDNA3.1-Lnc-Alt-G) into SK-N-SH cells, quantitative PCR (qPCR) was performed to analyze the expression of specific RNA. All transfections were carried out using Lipofectamine2000 Transfection Reagent (Invitrogen) according to the manufacturer's instructions.

## Viral constructs

The recombinant lentiviral vectors GV502-Ubi-Lnc-Ref-C-SV40-EGFP and GV502-Ubi-Lnc-Alt-G-SV40-EGFP were constructed by inserting full-length cDNA sequences of lncRNA *LINC02449*, representing different allelic genotypes, into the GV502-SV40-EGFP vector (ctrl) at the BamHI and AgeI sites. The lentiviruses were obtained from Shanghai Genechem Co., Ltd. Similarly, recombinant adeno-associated virus (AAV) vectors AAV-hsyn-Lnc-Ref-C-CMV-EGFP and AAV-hsyn-Lnc-Alt-G-CMV-EGFP were created by inserting the full-length lncRNA

*LINCO2449* with different allelic genotypes into the AAV-hsyn-EGFP vector (ctrl). The generated AAV were produced by BrainVTA (Wuhan).

## Quantitative real-time PCR (qRT-PCR)

Total RNA was extracted from the cells or tissues using Trizol (Invitrogen) according to the manufacturer's instructions, and then used for cDNA synthesis, followed by quantitative real-time polymerase chain reaction (qRT-PCR) using 2 × Hieff™ qPCR SYBR™ Green Master Mix (YEASEN, China) to quantify cDNA levels. The primers used in this study were as follows:

(1) Lnc-qPCR-F: 5'-ACAAGTAGCCATCCACCACA-3';
(2) Lnc-qPCR-R: 5'-AGACTGTGAAGCATCCTGTG-3';
(3) *CPLX1*-human-qPCR-F: 5'-CGCCATGGAGTTTGTGATGA-3';
(4) *CPLX1*-human-qPCR-R: 5'-CCTTCTTCTTGATGCCGTACTT-3';
(5) *Cplx1*-mice-qPCR-F: 5'-GGAACCAAGCCATCACCATG-3';
(6) *Cplx1*-mice-qPCR-R: 5'-TCCTTCTTCTTGATGCCATACTT-3'.
(7) *STMN2*-human-qPCR-F: 5'-GCTCTTGCTTTTACCCGGAAC-3';
(8) *STMN2*-human-qPCR-R: 5'-AGGCACGTTTGTTGATTTGCT-3';
(9) *Stmn2*-mice-qPCR-F: 5'-CAGAGGAGCGAAGAAAGTCTCA-3';
(10) *Stmn2*-mice-qPCR-R: 5'-CTAGATTAGCCTCACGGTTTTCC-3'.

Gene expression levels were normalized utilizing the $2^{-\Delta\Delta CT}$ method, with *Gapdh* as the reference gene. Data from three technical replicates were used for analysis.

## Animals and Stereotactic surgery

Experiments were conducted exclusively on male C57BL/6 mice (6–8 weeks old), obtained from the Laboratory Animal Center of Southern Medical University. Four to five C57BL/6 J mice were housed in an EVC cage (300 × 170 × 120 mm) at 23 ± 1 °C, humidity 40% under standard laboratory conditions with a 12 h light/dark cycle (lights on from 8:00 a.m. to 8:00 p.m.) and with free access to food and water. All procedures were approved by the Institutional Animal Care and Use Committee and the Ethics Committee of Guangdong General Hospital of Southern Medical University (#KY2024-915-02).

Mice were anesthetized with pentobarbital sodium (50 mg/kg) and placed in a stereotactic frame for surgery. A microsyringe containing the recombinant adeno-associated virus (AAV) was used to inject the virus into the medial prefrontal cortex (mPFC) at the following coordinates: AP = + 1.75 mm, ML = ± 0.34 mm, DV = − 2.75 mm. A total of 0.8 μL of virus was delivered over 4 min using a 20–40 psi nitrogen pulse, with the needle remaining in place for 3 min postinjection. Bilateral injections were performed for non-optogenetic experiments.

For optogenetic experiments, after AAV-CaMKIIa-hChR2(H134R) or AAV-CaMKIIa-EYFP injection into the unilateral mPFC, an optical fiber (length = 5.0 mm, diameter = 2.5 mm) was implanted into the NAc at AP = + 1.65 mm, ML = + 0.75 mm, DV = − 4.65 mm. The optical fiber was left in place for at least 3 min after implantation. Only mice with confirmed infection were included in the study.

## Behavioral experiments

Behavioral experiments were conducted on day 21 after AAV injection using a battery of assays, including the three-chamber social interaction test, grooming and marble burying test, open field test, elevated plus maze, sucrose preference test, Y-Maze, forced swim test, and tail suspension test. Mice were acclimatized to the behavioral chamber for 30 min before testing, with a minimum interval of 3 days between different experiments to avoid carryover effects. The methods used were based on previously published protocols[71–74].

**Three-chamber social preference test.** The three-chamber apparatus consisted of a plexiglass box (50 × 25 cm) divided into three compartments with a 5 cm aperture between each, and a movable partition door. The test mouse was placed in the central chamber during the habituation phase, with two empty wire cages in the adjacent chambers. After 10 min of exploration, a C57BL/6 mouse (S1), unfamiliar to the experimental mouse, was placed in one of the wire cages, while the other remained empty (E). The test mouse was allowed to explore for 10 min during the social exploration phase, followed by a social discrimination phase where a different C57BL/6 mouse (S2) was placed in the second cage. The duration of nose-to-nose interactions was tracked to assess social contact.

**Self-grooming test.** Mice were placed in a transparent acrylic box (30 × 30 × 35 cm) and allowed to explore freely for 10 min. The total time spent on spontaneous self-grooming behavior during this period was recorded.

**Marble burying test.** Mice were placed in a clean cage (46 × 20 × 14 cm) lined with at least 4 cm of sawdust bedding. Twenty black glass marbles (16 mm in diameter) were placed in a 4 × 5 pattern on the bedding. The mice were allowed to explore the cage for 30 minutes, after which the number of marbles buried was counted. A marble was considered buried if the surface area covered by sawdust exceeded 50% of the marble's surface area.

**Open field test**[75,76]. Mice were allowed to freely explore a square open-field arena (40 × 40 × 40 cm) for 30 min. To minimize external influence, experimenters exited the testing room during the exploration period, ensuring the mice were neither seen nor heard. After each trial, the arena was thoroughly cleaned with 75% ethanol to eliminate residual olfactory cues. The total distance traveled and the time spent in the central area were recorded using the Omnitech Open Field Monitoring System.

**Elevated plus-maze**[76,77]. The elevated plus maze consisted of two open arms and two closed arms (each 5 cm wide × 30 cm long), arranged in a cross-shaped configuration and elevated 50 cm above the floor. Mice were placed at the center of the maze facing an open arm and allowed to explore freely for 5 min. Behavior was recorded using an overhead camera connected to tracking software, which quantified the proportion of time spent in the open arms as an index of anxiety-like behavior.

**Sucrose preference test**[78]. Mice were individually housed in cages equipped with two drinking bottles. For the first two days, both bottles contained either water or 1% sucrose solution to establish baseline preference, with bottle positions swapped daily to eliminate positional bias. Following this acclimation phase, mice were randomly assigned one bottle of water and one bottle of 1% sucrose solution. Sucrose preference was assessed over three consecutive days. Bottle weights were recorded every 24 h to measure fluid consumption. Sucrose preference (%) was calculated as: [Sucrose intake / (Sucrose + Water intake)] × 100.

**Y-maze test**[79]. The Y-maze consisted of three arms, each marked with distinct wall patterns to facilitate spatial recognition. During the training session, one arm (designated as the novel arm) was blocked with an opaque barrier. Mice were placed at the start arm, facing the maze center, and allowed to explore the two accessible arms freely for 5 minutes. After a 10 min inter-trial interval, mice were reintroduced to the maze in the same position, but with the barrier removed, allowing access to all three arms. The test session lasted 5 min, and the percentage of time spent in the novel arm was recorded as a measure of spatial working memory. The maze was cleaned with 75% ethanol after each trial to eliminate olfactory cues.

**Forced swimming test**[77]. The test was conducted using a transparent Plexiglas cylinder (80 cm in height, 25 cm in diameter) filled with water

(22–24 °C) to a depth of 40 cm, preventing mice from touching the bottom or escaping. Each mouse was placed in the water for a 1 min habituation period (adaptation session), followed by a 5 min test session during which behavior was recorded. Immobility was defined as the absence of active swimming, with the mouse remaining afloat and making only minimal movements to maintain balance, such as slight limb or tail motions. The total duration of immobility was quantified as an index of behavioral despair.

**Tail suspension test**[80]. Each mouse was suspended by the tail inside an enclosed white box using a custom-made tail-fixation device to prevent escape. The test lasted for 6 min, during which the total duration of immobility was manually recorded. Mice were considered immobile when they ceased all active movements for at least one second, excluding passive swaying or natural shaking. Immobility time was used as an indicator of behavioral despair.

**Optogenetics.** Three weeks before the initiation of the three-compartment social preference test, experimental mice underwent unilateral virus injections targeting the medial prefrontal cortex (mPFC) and fiber optic implantation into the ipsilateral nucleus accumbens (NAc). The implanted fiber was connected to a laser emitting light at a wavelength of 470 nm for optogenetic activation. During the initial phase of the experiment, the test mice were placed in the central chamber of the three-chamber apparatus and allowed to explore freely for 5 min without optogenetic activation. Following this, the mice were reintroduced into the central chamber and allowed to explore for an additional 5 min with optogenetic activation. The time spent by the mice in front of each wire cage was recorded under both conditions. In the social exploration phase, a C57BL/6 mouse (S1) was randomly placed in one of the wire cages, and the same procedure was repeated. This design enabled the assessment of social interaction behaviors with and without optogenetic activation.

### Electrophysiological recordings

Mice were anesthetized with 1% sodium pentobarbital, followed by administration of 10 mL of chilled sucrose cutting solution to facilitate brain extraction. The sucrose solution consisted of (in mM) 228 sucrose, 26 NaHCO3, 11 glucose, 2.5 KCl, 1 NaH2PO4·H2O, 0.5 CaCl2·2H2O, and 7 MgSO4·7H2O. After euthanizing the mice via cervical dislocation, their brains were immediately immersed in the sucrose solution, and 300 μm thick slices were obtained using a Lycra LS1200s vibrating microtome. Slices containing the mPFC and NAc were selected and incubated in artificial cerebrospinal fluid (aCSF) at 35 °C for 30 min. The aCSF consisted of (in mM) 119 NaCl, 26 NaHCO3, 11 glucose, 2.5 KCl, 1 NaH2PO4·H2O, 2.5 CaCl2·2H2O, and 1.3 MgSO4·7H2O to properly prepare the tissue for further experimentation.

Following the 30 min incubation, the slices were allowed to equilibrate to room temperature for one hour. The brain slices were then transferred to an immersion recording chamber and continuously perfused with oxygenated aCSF (95% O2/5% CO2) at room temperature, at a flow rate of 3 mL/min. Neurons within the mPFC were visualized using a 40 × water immersion objective on a light microscope. For electrophysiological recordings, a recording electrode was created from a borosilicate capillary glass tube with a resistance of 4–8 MΩ using a P97 electrode puller. The electrode was filled with a solution containing (in mM) 120 CsMeSO4, 15 CsCl, 10 TEA-Cl, 8 NaCl, 10 HEPES, 5 QX-314, 4 ATP-Mg, 1 EGTA, and 0.3 GTP-Na. This electrode was inserted into the cell membrane to measure action potential characteristics through step current injections. To measure excitatory synaptic activity, locally induced excitatory postsynaptic currents (EPSCs) were stimulated at a frequency of 0.05 Hz using bipolar electrical stimulation electrodes. Miniature excitatory postsynaptic currents (mEPSCs) were recorded in the presence of 1 μM tetrodotoxin

(Aladdin) and 20 μM bicuculline (Sigma, USA), which block sodium channels and GABA receptors, respectively.

### FM 4-64 imaging

To assess vesicle release and endocytic trafficking, we performed FM4-64 dye unloading assays in SK-N-SH and SH-SY5Y cells stably over-expressing the *LINCO2449* Ref-C and Alt-G, and in SK-N-SH cells transiently transfection withe either pLKO.1-sh*Cplx1* or pLKO.1-shCtrl vectors. Cells were plated on confocal dishes and pre-incubated in saline solution for at least 10 min. FM4-64 (10 μM; Invitrogen™) was applied for 2 min in a depolarizing high-K$^+$ solution (75 mM KCl in a solution containing in mM: 170 NaCl, 3.5 KCl, 0.4 KH2PO4, 5 NaHCO3, 1.2 Na2SO4, 1.2 MgCl2, 1.3 CaCl2, 5 glucose, and 20 N-tris(hydroxymethyl)-methyl-2-aminoethane-sulfonic acid; pH 7.4). After stimulation, cells were rinsed and incubated for 10 min in FM4-64-cotaining saline solution, followed by a 5 min perfusion and 10 min incubation in dye-free saline solution to allow dye unloading.

FM 4-64 imaging was performed on a Zeiss LSM 880 Airyscan confocal microscope equipped with a C-Apochromat 40 × /1.2 W Korr FCS M27 objective. Time-lapse images were acquired every 1.26 s at 25 °C. During acquisition, cells were stimulated with 75 mM KCl saline for 8 min. Excitation was set to 488 nm, and emission was collected at 562 nm. Fluorescence intensity was quantified using the formula:

$$F = (F1/(F0 − B0))$$

where $F1$ in the fluorescence intensity at each time point, $F0$ is the baseline fluorescence intensity, and $B0$ is background fluorescence. Data were normalized to the baseline mean fluorescence intensity, as described previously[81]. At least 10 cells per condition were analyzed across a minimum of two independent experiments.

### Chromatin Isolation by RNA Purification (ChIRP)

We employed LongTarget to predict potential *LINCO2449* binding regions in the *CPLX1* gene[47] and performed Chromatin Isolation by RNA Purification (ChIRP) using the ChIRP Kit (BersinBio, Guangzhou, China) to validate the interaction between *LINCO2449* and *CPLX1*. SK-N-SH neuroblastoma cells stably expressing the *LINCO2449* alternative allele (via lentiviral transduction) were harvested and crosslinked with 1% formaldehyde for 20 minutes at room temperature. Crosslinking was quenched with 0.125 M glycine for 5 min. Cells were then lysed in ChIRP lysis buffer, and chromatin was sheared by sonication into ~100–500 bp fragments.

After pre-clearing, lysates were incubated at 37 °C for 3 h with biotinylated *LINCO2449*-specific probes in two separate reactions: one containing odd-number probes (#1, 3, 5) and the other containing even-numbered probes (#2, 4, 6). Following capture, the DNA was purified and analyzed by qPCR using primers (Forward primer: GATGGCAACAAGAACCCTGC; Reverse primer: GGCCTGGGTGACATCAAGTT) spanning the predicted *LINCO2449* binding region in the upstream region of *CPLX1*. *GAPDH* was included as a negative control target (Forward primer: ATTTGGCTACAGCAACAGGGT; Reverse primer: GAGGGGAGATTCAGTGTGGTG).

### Statistical analysis

Statistical analyses were conducted using GraphPad Prism 9.0 or SPSS software. Pearson correlation analysis, two-tailed Student's *t* test, and ANOVA were performed, as appropriate. A significance level of $P < 0.05$ was considered statistically significant.

### Reporting summary

Further information on research design is available in the Nature Portfolio Reporting Summary linked to this article.

## Data availability
The RNA-seq data generated in this study (including quantification and raw sequencing files) have been deposited in the GEO database under the accession codes GSE295729 (for SK-N-SH cells; https://www.ncbi.nlm.nih.gov/geo/query/acc.cgi?acc=GSE296023), and GSE296023 (for murine brain tissues; https://www.ncbi.nlm.nih.gov/geo/query/acc.cgi?acc=GSE296023). All data supporting the findings described in this manuscript are available in the article and in the Supplementary Information. Source data are provided in this paper.

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

## Acknowledgements

We thank the National Natural Science Foundation of China [grant number 82101577 to ZW, 82471527 to CZ], China Postdoctoral Science Foundation [2020M682806 to ZW], Guangdong-Hong Kong Joint Laboratory for Psychiatric Disorders [grant number 2023B1212120004 to XC], and the Guangdong Science and Technology Foundation [grant number 2019B030316032 to CZ] for providing financial supports.

## Author contributions

Conceived and designed the experiments: T.Y., J.L., Z.W., and C.Z. Performed the experiments: T.Y., J.L., Z.D., Q.C., Y.W., S.L., Y.L., C.N., and Z.W. Analyzed the data: T.Y., J.L., H.X., Z.W., and C.Z. Collected and diagnosed the control and patient subjects: T.J. and MJ. Wrote the paper: T.Y., J.L., Z.D., Z.W., X.C., and C.Z.

## Competing interests

The authors declare no competing interests.

## Additional information

[1]Key Laboratory of Mental Health of the Ministry of Education, Guangdong-Hong Kong-Macao Greater Bay Area Center for Brain Science and Brain-Inspired Intelligence, Guangdong-Hong Kong Joint Laboratory for Psychiatric Disorders, Guangdong Province Key Laboratory of Psychiatric Disorders, Guangdong Basic Research Center of Excellence for Integrated Traditional and Western Medicine for Qingzhi Diseases, and School of Basic Medical Sciences, Southern Medical University, Guangzhou, China. [2]Department of Medical Genetics, Guangdong Technology and Engineering Research Center for Molecular Diagnostics of Human Genetic Diseases, Guangdong Engineering and Technology Research Center for Genetic Testing, and Experimental Education/Administration Center, School of Basic Medical Sciences, Southern Medical University, Guangzhou, China. [3]Department of Neurobiology, School of Basic Medical Sciences, Southern Medical University, Guangzhou, China. [4]The Third People's Hospital of Zhongshan, Zhongshan, Guangdong, China. [5]Guangdong Mental Health Center, Guangdong Provincial People's Hospital (Guangdong Academy of Medical Sciences), Southern Medical University, Guangzhou, China. [6]Division of Life Science, The Hong Kong University of Science and Technology, Clear Water Bay, Kowloon, Hong Kong, China. [7]These authors contributed equally: Tengfei Yang, Jin-Ming Liu, Qiaqi Chen, Zhiying Deng, Chaoying Ni. ✉e-mail: caoxiong@smu.edu.cn; zhongjuwang315@163.com; cyzhao@smu.edu.cn

