## [Transparent Peer Review file · Nature Communications]

Gain of Alternative Allele Expression of LINC02449 at rs149707223 in Schizophrenia and Bipolar Disorder: Inducing Synaptic Transmission and Behavioral Deficits in Mice

Corresponding Author: Professor Cunyou Zhao

Version 0:

Reviewer comments:

Reviewer #1

(Remarks to the Author)

1. The abstract uses "social desirability deficits," which may cause confusion. "Social interaction deficits" is more accurate and consistent with the main text. Please revise accordingly.
2. In lines 129–132, the authors report selecting ASE-lncRNAs using a Bayes factor (BF) > 5 threshold and further filtering based on differential expression data from PsychENCODE. However, the manuscript provides limited information supporting this filtering step. Figure 1B illustrates transitions in ASE patterns between unaffected and affected individuals but does not directly substantiate the integration or results of PsychENCODE differential expression data. Additional details or supplementary data should be provided to justify this filtering criterion.
3. In lines 145–147, the statement that LINC02449 "showed a trend in SCZ patients" is not supported by the reported values ($\log_2FC = 0.002$, $P = 0.77$). This characterization is misleading and should be revised to accurately reflect the lack of statistical significance.
4. The study emphasizes the mPFC–NAc pathway but does not explore other relevant circuits (e.g., hippocampus, amygdala). Please briefly acknowledge this in the discussion and suggest directions for future work.
5. The legend for Figure 2 lacks sufficient detail on statistical methods and results. The text in lines 703–719 specifies one-way ANOVA for all panels, but this method may not be appropriate for Figure 2E, depending on its design. Moreover, P-values alone are insufficient; the authors should report descriptive statistics (e.g., means, standard deviations, confidence intervals, or effect sizes) to provide a fuller understanding of the observed effects.
6. Supplementary Figure 2 lacks a description of the statistical analyses performed. The authors should specify the statistical tests used, the number of replicates or samples, and the corresponding results, including P-values and summary statistics.
7. In lines 192–194, the authors refer to an adjustment procedure but do not specify which method was used (e.g., Bonferroni, Benjamini-Hochberg FDR), which should be clearly stated.
8. The methods section omits citations for several analytical approaches and tools. Key procedures—such as ASE detection, RNA-seq processing, and statistical modeling—should be accompanied by appropriate references to ensure reproducibility and transparency.
9. The discussion section would benefit from a critical evaluation of the study's limitations. Potential issues such as sample size, generalizability, or tissue specificity of ASE should be acknowledged to provide a balanced interpretation of the findings and guide future research directions.

Reviewer #2

(Remarks to the Author)

This study presents a compelling investigation into ASE as a molecular mechanism linking noncoding regulatory variants to SCZ and BD. The use of phenotypically discordant monozygotic (MZ) twins is a particularly elegant design, as it minimizes genetic background noise and enhances the power to detect environment- or epigenetic-driven expression differences. By identifying functional ASE variants and validating downstream regulatory effects using transcriptomic and behavioral data, this work provides important mechanistic insights into psychiatric disease vulnerability, with clear scientific merit and translational relevance. Several aspects of the writing and methodological detail require clarification or improvement to fully support the research conclusions.

1. In lines 39–40, the abstract describes BDSCZ as a complex polygenic disorder characterized by significant phenotypic

variability. However, the manuscript does not explore intra-diagnostic phenotypic variation; rather, it compares affected individuals to healthy controls. The description should be revised for precision to prevent misinterpretation of the study's scope.

2. The introduction would benefit from the inclusion of additional references to support key claims. Specifically, citations should be added in lines 67–68 regarding GWAS in psychiatric disorders, and in lines 74–75 to substantiate the tissue specificity of ASE.
3. While CPLX1 is identified as a target of LINC02449, the relevance and importance of CPLX1 should be more described, and other potential downstream pathways are also introduced. Please consider expanding the discussion to include broader regulatory networks or synergistic mechanisms.
4. The discussion should be reinforced with additional literature that contextualizes the findings, particularly regarding the functional impact of SNPs within lncRNAs. Referencing recent studies that demonstrate how regulatory variants influence lncRNA expression and function would strengthen the interpretation and significance of the results.
5. In Figure 5D, the label "Ctrl" should be made consistent with the labeling used for the "KCI" treatment group. Harmonizing terminology across all figure panels will improve clarity and professionalism.
6. Attention to stylistic consistency is recommended. For instance, in Figure 6, asterisks denoting statistical significance appear in both bold and non-bold formats. Additionally, p-values are inconsistently presented (e.g., both uppercase and lowercase italics), and LINC02449 appears in both italicized and non-italicized styles. A systematic review of text, figures, and tables is advised to ensure uniform formatting throughout the manuscript.
7. Line 161-194 In manuscript "LINC02449 expression was upregulated in the cortex in the PsychENCODE dataset and was also highly expressed in the human cortex according to the GTEx dataset ", the detailed results should be presented in the main text or supplementary files.

Reviewer #3

(Remarks to the Author)

This study "Gain of Alternative Allele Expression of LINC02449 at rs149707223 in Schizophrenia and Bipolar Disorder: Inducing Synaptic Transmission and Behavioral Deficits in Mice" investigates the role of allele-specific expression (ASE) of long non-coding RNAs (lncRNAs) in the pathogenesis of schizophrenia (SZ) and bipolar disorder (BD) using phenotype-discordant monozygotic (MZ) twins. The researchers identified a significant ASE shift in LINC02449 at SNP rs149707223 (C/G), with the alternative G allele predominating in affected individuals, contrasting with the reference C allele in unaffected twins. This shift was associated with increased LINC02449 expression in BD and a trend in SZ in the PsychENCODE brain RNA-seq dataset. Further, experiments in mice showed that the rs149707223 alternative allele drives overexpression of LINC02449, resulting in social behavior deficits and repetitive behaviors. These abnormalities were linked to enhanced CPLX1-mediated excitatory synaptic transmission in the mPFC–NAc circuit, implicating allele-specific expression (ASE) of LINC02449 as an unstudied regulatory mechanism in BD and SCZ. This is an interesting paper, however I had a few concerns:

Major concerns:

1. The paper verifies CPLX1 as a target of LINC02449 through integrative genomic analysis, expression studies, functional assays, and rescue experiments, providing strong evidence of causality rather than just correlation. However, these questions need to be addressed:
 - The lack of direct interaction evidence poses a limitation. The discussion acknowledges this limitation, suggesting future studies to explore the detailed interaction mechanisms (Page 12, Lines 338-339) which raises the question whether CPLX1 is a mediator or a direct target?
 - CPLX1 is a major regulator of synaptic vesicle exocytosis, so its knockdown might broadly affect synaptic transmission, masking the effects of other LINC02449 targets. The authors did not test whether knocking down other co-expressed genes (e.g., the other four overlapping DEGs in Fig. 5A) also rescues the phenotype? These genes have been reported to be linked to psychiatric and neurological disorders.
 - The study uses shRNA to knock down Cplx1, but off-target effects of shRNA can confound results. The authors provide a control shRNA sequence (Page 17, Lines 473-474), but they don't report whether this control was used in vivo to rule out non-specific effects of shRNA delivery. Additionally, the efficiency and specificity of Cplx1 knockdown (e.g., off-target gene silencing) are not fully detailed beyond qPCR and Western blotting (Fig. 6C, D).
2. The authors performed a range of behavioral tests in mice but did not adequately capture mania, depression, broader cognitive impairments, or psychotic-like symptoms—key features of SZ and BD. While social deficits and repetitive behaviors are relevant to both disorders, the absence of these other phenotypes means the model does not fully represent SZ/BD. A more comprehensive behavioral testing battery, including assays for mania and psychosis, would be needed to support such a claim. The model is better described as a targeted model for specific SZ/BD-related phenotypes rather than a general model for these disorders.
3. The authors suggest ASE as a therapeutic target, but the study lacks discussion on what are the practical challenges of targeting nuclear lncRNAs or synaptic proteins like CPLX1 in the brain?
4. The findings link LINC02449 ASE to SZ/BD susceptibility, but how might this translate to clinical diagnostics or risk stratification, especially since ASE is dynamic and tissue-specific?
5. Could this ASE shift (G allele in affected twins, C allele in unaffected) be influenced by environmental factors or medication effects in affected twins, given their disease status? What could be the potential reason for finding such similarity

in monozygotic twins from two distinct neuropsychiatric disorders? Please discuss

6. The authors did not report subtypes of BD such as type I, II, lithium responders? non-responders? etc.

Minor

o The number of replicates for vesicles release assay (Fig 5D) in SK-N-SKH cells are missing . Details could not be found in the materials and methods section?

o The study uses multiple datasets (PsychENCODE, GTEx, LIBD DLPFC, BrainSeq), but it's unclear if raw sequencing data (WGS, RNA-seq) from the twins or mouse experiments are publicly available. Will these be deposited in a repository for independent validation?

o How was the statistical power calculated for detecting ASE differences? Could a larger sample size reveal additional ASE-SNPs or reduce false positives, given the heterogeneity of SZ and BD? Please explain

Version 1:

Reviewer comments:

Reviewer #1

(Remarks to the Author)

Reviewer #2

(Remarks to the Author)

The authors have addressed my previous comments.

Reviewer #3

(Remarks to the Author)

The authors have addressed all of my concerns.

Dear Reviewers,

Re: NCOMMS-25-26100

Gain of Alternative Allele Expression of LINC02449 at rs149707223 in Schizophrenia and Bipolar Disorder: Inducing Synaptic Transmission and Behavioral Deficits in Mice

We thank the Reviewers for their constructive comments on the captioned manuscript. In response, we have carefully addressed each point and revised the manuscript accordingly. All changes have been highlighted in yellow in the revised version for ease of reference.

Differences in figures and tables between the revised and original versions

Revised version	Original version
Figures 1-4	Figures 1-4
Figure 5A (Modified), B (Modified), C	Figure 5A, B, C
Figure 5D (New)	
Figure 6A, B	Figure S5
Figure 6C	Figure 5D
Figure 7	Figure 6
Figure S1	Figure S1
Figure S2 (New)	
Figure S3 A-F	Figure S2 A-F
Figure S3 G, H (New)	
Figures S4,5	Figures S3, 4
Figure S6 (New)	
Figure S7 (New)	
Table S1	Table S1
Table S2 (New)	
Tables S3-8	Tables S2-7

Reviewer #1:

1.1 The abstract uses "social desirability deficits," which may cause confusion. "Social interaction deficits" is more accurate and consistent with the main text. Please revise accordingly.

Response: We thank the Reviewer for the valuable comments. In response, we have replaced "social desirability deficits" with "social interaction deficits" throughout the manuscript.

1.2 In lines 129–132, the authors report selecting ASE-lncRNAs using a Bayes factor (BF) > 5 threshold and further filtering based on differential expression data from PsychENCODE. However, the manuscript provides limited information supporting this filtering step. Figure 1B illustrates transitions in ASE patterns between unaffected and affected individuals but does not directly substantiate the integration or results of PsychENCODE differential expression data. Additional details or supplementary data should be provided to justify this filtering criterion.

Response: In response to the Reviewer's comments, we have now included a new **Supplementary Table 2**, which provides detailed information for the 15 lncRNA ASE sites. This table includes ASE data from twin samples as well as corresponding DEG information from the PsychENCODE dataset.

1.3 In lines 145–147, the statement that LINC02449 "showed a trend in SCZ patients" is not supported by the reported values ($\log_2FC = 0.002$, $P = 0.77$). This characterization is misleading and should be revised to accurately reflect the lack of statistical significance.

Response: We thank the Reviewer for the insightful comments, and we have now revised the description accordingly, as shown in Lines 155-158: "In line with this, LINC02449 expression was observed to be elevated significantly in BD patients ($\log_2FC=0.1408$, $P=0.002$; **Supplementary Table 2**) and showed a non-significant trend toward upregulation in SZ patients ($\log_2FC=0.009$, $P=0.770$) in the PsychENCODE brain RNA-seq dataset"

1.4 The study emphasizes the mPFC–NAc pathway but does not explore other relevant circuits (e.g., hippocampus, amygdala). Please briefly acknowledge this in the discussion and suggest directions for future work.

Response: We appreciate the Reviewer's insightful suggestion, and we fully agree that other circuits such as the hippocampus and amygdala may also be involved in mediating the effects of LINC02449 and CPLX1 on psychiatric phenotypes. We have now added a statement in the Discussion section (Lines 368-347): "Moreover, while our results highlight the critical involvement of the mPFC–NAc circuit in mediating the observed behavioral and synaptic alterations, we recognize that other neural regions and circuits implicated in SZ and BD pathophysiology, such as the hippocampus, amygdala, and thalamus, were not assessed. Future investigations utilizing region-specific manipulations and comprehensive circuit mapping will be important to clarify whether LINC02449 influences broader neural networks relevant to psychiatric disorders."

1.5 The legend for Figure 2 lacks sufficient detail on statistical methods and results. The text in lines 703–719 specifies one-way ANOVA for all panels, but this method may not be appropriate for Figure 2E, depending on its design. Moreover, P-values alone are insufficient; the authors should report descriptive statistics (e.g., means, standard deviations, confidence intervals, or effect sizes) to provide a fuller understanding of the observed effects.

Response: In response to the Reviewer’s concerns, we have now provided the detailed descriptive statistics in Lines 1001-1003: “All data represent as means \pm standard errors of the mean (SEMs). One-way ANOVA followed by Bonferroni’s multiple comparison test was conducted for the indicated comparisons, or Student’s t-test was used for comparisons between two groups.”

1.6 Supplementary Figure 2 lacks a description of the statistical analyses performed. The authors should specify the statistical tests used, the number of replicates or samples, and the corresponding results, including P-values and summary statistics.

Response: In response to the Reviewer’s concerns, we have now provided the detailed descriptive statistics for **Supplementary figure 3** (Supplementary figure 2 in previous version):” Data points for individual mice are shown using distinct symbol (Δ , \circ , ∇) and presented as mean \pm SEM. One-way ANOVA was conducted for the indicated comparisons. n.s., not significant”

1.7 In lines 192–194, the authors refer to an adjustment procedure but do not specify which method was used (e.g., Bonferroni, Benjamini-Hochberg FDR), which should be clearly stated.

Response: We thank the Reviewer for this important observation. We have now specified that the Benjamini-Hochberg false discovery rate (FDR) correction was applied to control for the false discovery rate. This clarification has been added to the revised manuscript (Line 217).

1.8 The methods section omits citations for several analytical approaches and tools. Key procedures—such as ASE detection, RNA-seq processing, and statistical modeling—should be accompanied by appropriate references to ensure reproducibility and transparency.

Response: We thank the Reviewer for this important observation. We have now carefully revised the Methods section to include appropriate citations for all key analytical approaches and tools used in this study. Specifically, we have:

- Added references for SNP identification, haplotype construction, and allele-specific expression (ASE) detection methods and software tools (e.g., BWA, GATK4, Annovar, SHAPEIT2, vcf2doploid, EMASE, and INLA as applicable);
- Cited standard RNA-seq processing tools such as hisat2 for alignment, stringtie and featureCounts for quantification, and DESeq2 for DEG analysis;
- Added reference for TDF, which was used to predict the potential for RNA:DNA triplex complex;
- Included references for statistical analyses, including the use of one-way ANOVA,

Benjamini-Hochberg FDR correction, and linear mixed models where relevant.

These revisions can be found in the updated Methods section (Lines 446–567). We appreciate the Reviewer’s attention to ensuring reproducibility and transparency.

1.9 The discussion section would benefit from a critical evaluation of the study’s limitations. Potential issues such as sample size, generalizability, or tissue specificity of ASE should be acknowledged to provide a balanced interpretation of the findings and guide future research directions.

Response: We thank the Reviewer for this valuable suggestion. In response, we have revised the Discussion section to explicitly acknowledge and critically evaluate several key limitations of our study. Specifically, we addressed issues related to the tissue specificity of ASE in Lines 413-417, and discuss sample size and generalizability in Lines 429-435. These additions provide a more balanced interpretation of our findings and highlight important directions for future research.

Reviewer #2:

This study presents a compelling investigation into ASE as a molecular mechanism linking noncoding regulatory variants to SCZ and BD. The use of phenotypically discordant monozygotic (MZ) twins is a particularly elegant design, as it minimizes genetic background noise and enhances the power to detect environment- or epigenetic-driven expression differences. By identifying functional ASE variants and validating downstream regulatory effects using transcriptomic and behavioral data, this work provides important mechanistic insights into psychiatric disease vulnerability, with clear scientific merit and translational relevance. Several aspects of the writing and methodological detail require clarification or improvement to fully support the research conclusions.

2.1 In lines 39–40, the abstract describes BDSCZ as a complex polygenic disorder characterized by significant phenotypic variability. However, the manuscript does not explore intra-diagnostic phenotypic variation; rather, it compares affected individuals to healthy controls. The description should be revised for precision to prevent misinterpretation of the study’s scope.

Response: We thank the Reviewer for pointing out this important issue. To improve precision and avoid potential misinterpretation, we have revised the sentence in the abstract to clarify that our focus was on identifying molecular differences between affected individuals and healthy controls, rather than exploring phenotypic variability within diagnostic categories in the Abstract of revised manuscript (Lines 42-47).

2.2 The introduction would benefit from the inclusion of additional references to support key claims. Specifically, citations should be added in lines 67–68 regarding GWAS in psychiatric disorders, and in lines 74–75 to substantiate the tissue specificity of ASE.

Response: We thank the Reviewer for this helpful suggestion. In response, we have revised the Introduction to include additional references that support the key claims identified:

- For **Lines 73-74**, we have cited several large-scale genome-wide association studies (GWAS) in psychiatric disorders, including studies from the Psychiatric Genomics Consortium that highlight the polygenic architecture of schizophrenia and bipolar disorder (e.g., Trubetskoy et al., 2022; O'Connell et al., 2025; Bipolar Disorder and Schizophrenia Working Group of the Psychiatric Genomics Consortium, 2018).
- For **Lines 80-82**, we have added references to studies demonstrating the tissue and cell-type specificity of allele-specific expression, particularly in brain tissue (e.g., Juan et al., 2022; Qi et al., 2025; Pierre et.al, 2022; Castel et al., 2020).

2.3 While CPLX1 is identified as a target of LINC02449, the relevance and importance of CPLX1 should be more described, and other potential downstream pathways are also introduced. Please consider expanding the discussion to include broader regulatory networks or synergistic mechanisms.

Response: We appreciate the Reviewer's insightful comment. In response, we have expanded the Discussion section to further elaborate on the functional relevance of CPLX1 in the context of psychiatric disorders in the revised Discussion section (**Lines 395-405**), and they help place our findings within a broader biological and regulatory context.

“Although CPLX1 was prioritized for functional rescue experiments due to its robust differential expression, positive correlation and direct interaction with LINC02449, as well as its well-established roles in synaptic regulation, we acknowledge that it may not be the sole relevant effector gene. Knockdown of Cplx1 could broadly affect synaptic transmission, potentially masking contributions of other genes co-expressed with LINC02449. For instance, STMN2 identified in Fig. 5A, exhibited consistent upregulation in RNA-seq analyses, although the increase did not reach statistical significance in qPCR validation. Notably, STMN2 has previously been implicated in neuronal growth regulation and Alzheimer's disease pathogenesis^{53,54}. Thus, future studies are warranted to explore whether targeted manipulation of STMN2— individually or in combinatorial with other candidate gene— can rescues the synaptic and behavioral phenotypes observed in our model.”

2.4 The discussion should be reinforced with additional literature that contextualizes the findings, particularly regarding the functional impact of SNPs within lncRNAs. Referencing recent studies that demonstrate how regulatory variants influence lncRNA expression and function would strengthen the interpretation and significance of the results.

Response: We thank the Reviewer for this constructive suggestion. In response, we have revised the Discussion (**Lines 331-338**) to incorporate recent literature that highlights the functional consequences of regulatory SNPs within lncRNAs. These references provide important context for understanding how non-coding variants can affect lncRNA expression, stability, subcellular localization, and interactions with chromatin or RNA-binding proteins.

“Recent studies have shown that SNPs within lncRNA loci can influence various aspects of lncRNA biology, including expression, subcellular localization, RNA stability, and interaction with chromatin-modifying complex or RNA-binding proteins. For example, SNPs in MIAT and LINC-PINT have been linked to altered expression and splicing patterns in

neuropsychiatric conditions. These findings align with our observation that that regulatory ASE-SNP within lncRNAs, exemplified by LINC02449, can significantly shape that gene regulatory networks critical for neuronal processes and neuropsychiatric disease phenotypes”

2.5 In Figure 5D, the label "Ctrl" should be made consistent with the labeling used for the "KCl" treatment group. Harmonizing terminology across all figure panels will improve clarity and professionalism.

Response: We thank the Reviewer for this helpful suggestion. To enhance clarity and ensure consistency, we have revised the labels in **Figure 6C (Figure 5D in the previous version)** of the revised manuscript as follows: “ShCtrl” for control shRNA; “ShCPLX1” for CPLX1 shRNA; “Alt+ShCtrl” for co-transfection of the alternative allele with control shRNA; “Ref+ShCtrl” for co-transfection of the reference allele with control shRNA, and “Alt+ShCPLX1” for co-transfection of the alternative allele with CPLX1 shRNA.

2.6 Attention to stylistic consistency is recommended. For instance, in Figure 6, asterisks denoting statistical significance appear in both bold and non-bold formats. Additionally, p-values are inconsistently presented (e.g., both uppercase and lowercase italics), and LINC02449 appears in both italicized and non-italicized styles. A systematic review of text, figures, and tables is advised to ensure uniform formatting throughout the manuscript.

Response: We appreciate the Reviewer’s attention to detail. In response, we have carefully reviewed the entire manuscript, including the main text, figures, and tables, to ensure consistency in formatting. Specifically:

- Asterisks indicating statistical significance in **Figure 7 (Figure 6 in the previous version)** and all other figures have been standardized in both font weight and style.
- All *P*-values are now consistently formatted using uppercase italic text (e.g., *P* < 0.05).
- Gene symbols, including *LINC02449*, have been consistently italicized throughout the manuscript in accordance with standard gene nomenclature guidelines.

2.7 Line 161-194 In manuscript "LINC02449 expression was upregulated in the cortex in the PsychENCODE dataset and was also highly expressed in the human cortex according to the GTEx dataset ", the detailed results should be presented in the main text or supplementary files.

Response: We thank the Reviewer for highlighting this omission. In response, we have now included detailed expression analysis results of *LINC02449* from the PsychENCODE dataset in **Supplementary Table 2** and from the GTEx datasets in **Supplementary Figure 2**. These additions have been referenced in the revised main text (**Lines 173-176**) for clarity and transparency.

Reviewer #3:

This study" Gain of Alternative Allele Expression of LINC02449 at rs149707223 in Schizophrenia and Bipolar Disorder: Inducing Synaptic Transmission and Behavioral Deficits in Mice" investigates the role of allele-specific expression (ASE) of long non-coding RNAs

(lncRNAs) in the pathogenesis of schizophrenia (SZ) and bipolar disorder (BD) using phenotype-discordant monozygotic (MZ) twins. The researchers identified a significant ASE shift in LINC02449 at SNP rs149707223 (C/G), with the alternative G allele predominating in affected individuals, contrasting with the reference C allele in unaffected twins. This shift was associated with increased LINC02449 expression in BD and a trend in SZ in the PsychENCODE brain RNA-seq dataset. Further, experiments in mice showed that the rs149707223 alternative allele drives overexpression of LINC02449, resulting in social behavior deficits and repetitive behaviors. These abnormalities were linked to enhanced CPLX1-mediated excitatory synaptic transmission in the mPFC–NAc circuit, implicating allele-specific expression (ASE) of LINC02449 as an unstudied regulatory mechanism in BD and SCZ. This is an interesting paper, however I had a few concerns:

Major concerns:

3.1 The paper verifies CPLX1 as a target of LINC02449 through integrative genomic analysis, expression studies, functional assays, and rescue experiments, providing strong evidence of causality rather than just correlation. However, these questions need to be addressed:

3.1.1 The lack of direct interaction evidence poses a limitation. The discussion acknowledges this limitation, suggesting future studies to explore the detailed interaction mechanisms (Page 12, Lines 338-339) which raises the question whether CPLX1 is a mediator or a direct target?

Response: We thank the Reviewer for this insightful comment and for acknowledging the strength of our multi-level evidence linking CPLX1 to *LINC02449* function. To directly address this question raised, we utilized LongTarget to predict potential *LINC02449* binding sites within the *CPLX1* upstream region (**Fig. S6**). Subsequently, we experimentally validated these predictions using a chromatin isolation by RNA purification (ChIRP) assay. Our ChIRP-qPCR results validated that *LINC02449* enrichments at the predicted region spanning from 5,572-5,650 bp upstream of the *CPLX1* transcription start site (**Fig. 5D**) shown as below, supporting a direct regulatory role of *LINC02449* in *CPLX1* transcription. This new experimental evidence has been incorporated into the revised manuscript (**Lines 285-291**).

3.1.2 CPLX1 is a major regulator of synaptic vesicle exocytosis, so its knockdown might broadly affect synaptic transmission, masking the effects of other *LINC02449* targets. The authors did not test whether knocking down other co-expressed genes (e.g., the other four

overlapping DEGs in Fig. 5A) also rescues the phenotype? These genes have been reported to be linked to psychiatric and neurological disorders.

Response: We thank the Reviewer raising this important point. In response, we refined our candidate-target selection pipeline using more stringent criteria as follows:

- (1) Fold change (FC)>1.5 and adjusted *P* value (Padj)<0.001 for SK-N-SH cells overexpressing *LINC02449*;
- (2) FC>1.2 and Padj<0.001 for mouse mPFC overexpressing *LINC02449*;
- (3) Positive correlation ($r>0.3$ and Padj<1.0e-5) with *LINC02449* expression in human PFC (GTEx dataset).

Applying these stringent criteria narrowed our candidate list to two genes—*CPLX1* and *STMN2*—both of which were significantly upregulated in *LINC02449*-expressing SK-N-SH cells and mouse mPFC, and positively correlated with *LINC02449* expression in the human mPFC brain (revised **Figure 5A** below; **Lines 268-271**).

Notably, an allele-specific increase in *CPLX1* expression driven by the *LINC02449* Alt G allele—but not in *STMN2*—was confirmed at the mRNA level by qPCR in SK-N-SH cells (**Figure 5B** above). Given the established role of *CPLX1* in synaptic transmission and neurobehavioral phenotypes, we prioritized this gene and subsequently examined whether its knockdown could rescue phenotypes induced by *LINC02449*-overexpression in cellular and mouse models. This clarification has been added to the Results section of the revised manuscript (**Lines 280-284**):

We have also added the following clarification to the Discussion section (**Lines 395-405**) of the revised manuscript to directly address the Reviewer’s concerns:

“Although *CPLX1* was prioritized for functional rescue experiments due to its robust differential expression, positive correlation and direct interaction with *LINC02449*, as well as its well-established roles in synaptic regulation, we acknowledge that it may not be the sole relevant effector gene. Knockdown of *Cplx1* could broadly affect synaptic transmission, potentially masking contributions of other genes co-expressed with *LINC02449*. For instance, *STMN2* identified in Fig. 5A, exhibited consistent upregulation in RNA-seq analyses, although the increase did not reach statistical significance in qPCR validation. Notably,

STMN2 has previously been implicated in neuronal growth regulation and Alzheimer's disease pathogenesis. Thus, future studies are warranted to explore whether targeted manipulation of STMN2— individually or in combinatorial with other candidate gene— can rescues the synaptic and behavioral phenotypes observed in our model.”

3.1.3 The study uses shRNA to knock down Cplx1, but off-target effects of shRNA can confound results. The authors provide a control shRNA sequence (Page 17, Lines 473-474), but they don't report whether this control was used *in vivo* to rule out non-specific effects of shRNA delivery. Additionally, the efficiency and specificity of Cplx1 knockdown (e.g., off-target gene silencing) are not fully detailed beyond qPCR and Western blotting (Fig. 6C, D).

Response: We thank the Reviewer for raising these important technical concerns regarding potential off-target effects associated with shRNA-mediated Cplx1 knockdown.

To mitigate non-specific effects of shRNA delivery, we employed the pLKO.1 vector containing a non-hairpin insert as a negative control, which has been widely validated as a non-targeting control. This construct was used in vesicle release assays (FM4-64) performed in human SK-N-SH cell lines (Fig. 6C). For *in vivo* experiments, an empty AAV vector was used as a control in AAV-infected mouse brain. These experimental controls are now clearly described in Figures 6C and 7A and have been detailed in the revised Methods (Lines 576-582).

Regarding the specificity of Cplx1 knockdown, we complemented our qPCR and Western blotting validation (Fig. 7C, D) with a targeted off-target analysis using BLAST-based algorithms, conducted in direct response to the Reviewer' comments. The top-ranked transcripts with sequence homology to the shRNA were identified and their expression levels were subsequently assessed by qPCR in mouse neuroblastoma N2a cell lines treated with shCplx1. No significant changes were observed for these potential off-target genes, as shown in Fig. S7 below, indicating high specificity of the shRNA construct. These methodological details have now been incorporated into the Methods section (Lines 583–589) of the revised manuscript.

3.2 The authors performed a range of behavioral tests in mice but did not adequately capture mania, depression, broader cognitive impairments, or psychotic-like symptoms—key

features of SZ and BD. While social deficits and repetitive behaviors are relevant to both disorders, the absence of these other phenotypes means the model does not fully represent SZ/BD. A more comprehensive behavioral testing battery, including assays for mania and psychosis, would be needed to support such a claim. The model is better described as a targeted model for specific SZ/BD-related phenotypes rather than a general model for these disorders.

Response: We appreciate the Reviewer's insightful comment. We agree that our current behavioral battery does not capture the full spectrum of symptomatology associated with SZ and BD. However, our assays were designed for target specific, translationally relevant behavioral domains that are frequently disrupted in these disorders. These include:

- Social interaction (three-chamber test),
- Stereotypic behaviors (grooming, marble burying),
- Anxiety-like behavior (elevated plus maze),
- Locomotor activity related to mania-like states (open-field test),
- Depressive-like behavior (forced swim test, tail suspension test, and sucrose preference test), and
- Spatial working memory (Y-maze)

This has now been clarified in the revised Results section (Lines 179-182): "Behavioral testing was conducted on day 21 after AAV injection using a battery of assays, including the three-chamber social interaction test, grooming and marble burying test, open field test, elevated plus maze, sucrose preference test, Y-Maze, forced swim test, and tail suspension test."

While these behavioral domains are indeed relevant to both disorders, we acknowledge that additional paradigms specifically probing mania (e.g., amphetamine-induced hyperlocomotion) and psychosis-like symptoms (e.g., latent inhibition or MK-801-induced behavioral responses) would be necessary to support broader claims regarding disease modeling.

To address this, we have added the following clarification to the Discussion section (Lines 363-368): " We acknowledge that the behavioral assays employed in this study do not fully capture the clinical heterogeneity observed in SZ and BD. Specifically, paradigms designed to assess mania-like states (e.g., amphetamine-induced hyperlocomotion) or psychosis-like behaviors (e.g., latent inhibition, prepulse inhibition under pharmacological challenge) were not included. Incorporating a more comprehensive behavioral battery in future studies will be essential to thoroughly evaluate the spectrum of SZ/BD-related phenotypes."

3.3 The authors suggest ASE as a therapeutic target, but the study lacks discussion on what are the practical challenges of targeting nuclear lncRNAs or synaptic proteins like CPLX1 in the brain?

Response: We thank the reviewer for this insightful comment. Indeed, while our findings highlight the pathological relevance of ASE in LINC02449 and its downstream effects on CPLX1-mediated synaptic transmission, we acknowledge the translational challenges in targeting such molecular entities, especially in the brain.

To address this concern, we have now included a paragraph in the Discussion section that outlines key practical limitations (Lines 406-413): “Our study implicates ASE in the pathogenesis of SZ and BD, particularly through the dysregulation of the LINC02449–CPLX1 axis. While CPLX1, a synaptic vesicle exocytosis regulator, represents a more tractable target, its essential role in neurotransmission raises concerns about off-target effects. In contrast, nuclear lncRNAs like LINC02449 are difficult to modulate due to their subcellular localization, structural ambiguity, and context-specific functions. Effective therapeutic targeting will require cell-type- and circuit-specific tools, potentially combined with allele-specific strategies to ensure precision and minimize adverse effects.”

We hope this addition provides a clearer perspective on the translational hurdles and emphasizes our cautious approach in interpreting the therapeutic potential of the identified targets

3.4 The findings link LINC02449 ASE to SZ/BD susceptibility, but how might this translate to clinical diagnostics or risk stratification, especially since ASE is dynamic and tissue-specific?

Response: We appreciate the reviewer’s thoughtful question regarding the translational value of ASE findings. Indeed, ASE is influenced by both genetic and epigenetic factors, and its dynamic and tissue-specific nature presents significant challenges for clinical application. Nevertheless, we believe our study offers several key implications that may support future diagnostic or risk stratification strategies. We have now added the following paragraph to the Discussion section (Lines 413-417): “From a diagnostic perspective, ASE patterns are often dynamic and tissue-specific, which can limit their utility as stable biomarkers. However, the consistent ASE shift favoring the LINC02449 Alt-G allele across SZ/BD-discordant MZ twin pairs– and its dysregulation in postmortem brain tissue–suggests that stable cis-regulatory ASE events may serve as informative biomarkers, particularly when integrated with transcriptomic and polygenic risk data.”

3.5 Could this ASE shift (G allele in affected twins, C allele in unaffected) be influenced by environmental factors or medication effects in affected twins, given their disease status? What could be the potential reason for finding such similarity in monozygotic twins from two distinct neuropsychiatric disorders? Please discuss

Response: We thank the reviewer for raising this critical point. Indeed, the influence of environmental exposures and medication on ASE is a key consideration in interpreting our findings, particularly in monozygotic (MZ) twins. To address this, we have added the following discussion to the Discussion section (Lines 418-423): “Despite potential influences from environmental or treatment, the concordant ASE pattern among genetically identical twins raised in similar early-life environments supports a genetically driven, disease-associated mechanism. Its presence in both SZ and BD further underscores a shared molecular pathology involving synaptic dysfunction. Together, these findings position LINC02449 ASE dysregulation as a convergent mechanism underlying SZ and BD, with implications for disease modeling precision diagnostics.”

We believe this addition appropriately addresses the concern and contextualizes the shared

ASE pattern in the broader framework of SZ/BD pathophysiology. We appreciate the reviewer's suggestion to deepen our interpretation.

3.6 The authors did not report subtypes of BD such as type I, II, lithium responders? non-responders? etc.

Response: We thank the Reviewer for raising this important point. We have clarified in the **Supplemental Table 1** of the revised manuscript that all affected individuals among BDC twins in our study were diagnosed as BD-I subtype and were lithium responders. Due to constraints in sample size and incomplete detailed clinical metadata across all twin pairs, we did not perform further stratification based on other clinical characteristics or treatment responses. We agree with the Reviewer that such stratification could provide valuable additional insights into the specificity and clinical relevance of ASE events, and we highlight this as an important direction for future research.

3.7 Minor

3.7.1 The number of replicates for vesicles release assay (Fig 5D) in SK-N-SKH cells are missing. Details could not be found in the materials and methods section?

Response: We thank the Reviewer for pointing out this omission. We have revised the **Figure 6** (Figs. 5D and S5 in the previous version) legend to clear specify the number of analyzed cells and biological replicates used in the vesicle release assay as follows (Lines 1063-1064): "At least 10 cells were analyzed per treatment condition across a minimum of two independent transfection experiments."

3.7.2 The study uses multiple datasets (PsychENCODE, GTEx, LIBD DLPFC, BrainSeq), but it's unclear if raw sequencing data (WGS, RNA-seq) from the twins or mouse experiments are publicly available. Will these be deposited in a repository for independent validation?

Response: We thank the Reviewer for highlighting this important issue regarding data accessibility and transparency. We have now clarified the availability of all datasets in the revised manuscript under the "**Data Availability**" section. Due to current regulations by the Chinese government, the human twin dataset cannot be publicly shared online. However, this dataset remains available upon reasonable request from qualified researchers, who may direct their inquiries to the corresponding authors. This has been clarified in the revised manuscript (Lines 775-780).

3.7.3 How was the statistical power calculated for detecting ASE differences? Could a larger sample size reveal additional ASE-SNPs or reduce false positives, given the heterogeneity of SZ and BD? Please explain

Response: We thank the reviewer for raising this important point. Our study design—focused on monozygotic (MZ) twins discordant for SZ or BD—was intended to reduce background genetic variability and enrich for functionally relevant ASE differences. However, we acknowledge the limitations in statistical power inherent in the relatively small sample size.

In response to the reviewer's question, we have added the following clarification to the Discussion sections (Lines 424-435): "Finally, due to the exploratory nature and limited availability of discordant MZ twin pairs, formal power calculations were not performed prior to data collection. Instead, we applied a stringent Bayesian factor ($BF > 5$) threshold within a generalized additive linear mixed model framework to robustly identify ASE differences while minimizing false positives. This approach incorporates twin pairing as a random effect and adjusts for overdispersion in allelic counts. Nevertheless, we recognize that a larger sample size could enhance the discovery of additional ASE-SNPs and improve the robustness of findings. Given the known heterogeneity in SZ and BD, expanding the cohort would likely uncover more condition-specific ASE events, reduce the risk errors, and improve the generalizability of the results. Future efforts should focus on validating these ASE patterns in larger case-control cohorts and integrating other omics layers, such as chromatin accessibility or methylation status, to refine the biological relevance of identified ASE-SNPs"

We hope our revisions meet with your approval.

Sincerely yours,

Cunyou ZHAO, PhD

Professor and Director of Department of Medical Genetics,
School of Basic Medical Sciences, Southern Medical University,
Guangzhou, Guangdong, China